# A/B/n Testing with Control in the Presence of Subpopulations

**Yoan Russac**
CNRS, Inria, ENS
Université PSL
yoan.russac@ens.fr

**Christina Katsimerou**
Booking.com
christina.katsimerou@booking.com

**Dennis Bohle**
Booking.com
dennis.bohle@booking.com

**Olivier Cappé**
CNRS, Inria, ENS
Université PSL
olivier.cappe@cnrs.fr

**Aurélien Garivier**
UMPA, CNRS
Inria, ENS Lyon
aurelien.garivier@ens-lyon.fr

**Wouter M. Koolen**
Centrum Wiskunde & Informatica
wmkoolen@cwi.nl

## Abstract

Motivated by A/B/n testing applications, we consider a finite set of distributions (called *arms*), one of which is treated as a *control*. We assume that the population is stratified into homogeneous subpopulations. At every time step, a subpopulation is sampled and an arm is chosen: the resulting observation is an independent draw from the arm conditioned on the subpopulation. The quality of each arm is assessed through a weighted combination of its subpopulation means. We propose a strategy for sequentially choosing one arm per time step so as to discover as fast as possible which arms, if any, have higher weighted expectation than the control. This strategy is shown to be asymptotically optimal in the following sense: if $\tau_\delta$ is the first time when the strategy ensures that it is able to output the correct answer with probability at least $1 - \delta$, then $\mathbb{E}[\tau_\delta]$ grows linearly with $\log(1/\delta)$ at the exact optimal rate. This rate is identified in the paper in three different settings: (1) when the experimenter does not observe the subpopulation information, (2) when the subpopulation of each sample is observed but not chosen, and (3) when the experimenter can select the subpopulation from which each response is sampled. We illustrate the efficiency of the proposed strategy with numerical simulations on synthetic and real data collected from an A/B/n experiment.

## 1 Introduction

A/B/n testing is a website optimization procedure where multiple versions of the content (called "arms" below) are compared, often in order to find the one with the highest conversion rate. However, many e-commerce companies use A/B/n testing not only to deploy the best product implementation, but primarily to draw post-experiment inferences [11]. The decision-making involves, besides experiment results, factors such as the cost of scaling-up a solution, external data, or whether the implementation fits in a broader theme. In this setting, each of the arms better than the default product (which we will refer to as the "control" arm) is a contender for being deployed and the interest is not only in the best arm.

35th Conference on Neural Information Processing Systems (NeurIPS 2021).

Given the control and $K \geq 1$ alternative implementations (variants), the simplest idea is to distribute the traffic uniformly among the arms; the arms that appear to be significantly better than the control at the end of the experiment are considered for deployment. While well-established, this process can be inefficient in terms of resources. Some alternatives are soon obviously worse (or better) than the control and would require fewer samples than the alternatives closer to the control. A second related shortcoming of the basic A/B/n testing approach is that setting the duration of the experiment –when done in advance– necessitates a very conservative approach by choosing a run-length that is sufficiently long to differentiate even the smallest possible changes.

To address these limitations, we consider in this work sequential testing policies that can both adjust the allocation of the samples and be stopped adaptively, in light of the data gathered during the experiment. In the terminology of multi-armed bandits, this corresponds to *pure exploration* problems (see, e.g., Chap. 33 of [15]). A pure exploration strategy will typically choose every minute (say), an allocation of traffic that favors arms for which the uncertainty is the highest. The experiment is stopped as soon as the significance is considered sufficient for every arm. Approaches have been developed in [8, 12, 10] for the identification of the single arm with the highest mean, a task called the Best Arm Identification (BAI) problem. In particular, [10] propose a strategy that is asymptotically optimal in the *fixed confidence setting*, meaning that, given a risk parameter $\delta$, it finds the best arm with probability at least $1 - \delta$, using an expected number of samples that is hardly improvable when $\delta$ is small. Later, [18] incorporated the special role of the control arm in BAI and proposed an algorithm that declares as winning arm the one with the highest mean only if it is significantly better than the control. In this paper, we propose a solution to the problem of identifying all the arms that are better than the control, in a framework that generalizes the fixed confidence setting. In order to provide useful tools for practical A/B/n testing, we address two additional issues.

First, traditional stochastic bandit models are based on the assumption that the arm samples are i.i.d., whereas real world data streams usually show trends or some form of inhomogeneity. A particular case of interest for website optimization are the seasonal patterns caused by time-of-day or day-of-week variations. We henceforth include in our model observed covariates (e.g. the time of the day, but possibly also the country of origin, or controlled covariates like the order in which partners appear on the page, etc.) that stratify the observations into homogeneous subpopulations. We study different scenarios, depending on how much interaction is possible with these subpopulations. We provide a sample complexity analysis and an efficient algorithm in each case. In particular, we will show that using the subpopulation information efficiently can provide significant speedups of the decision-making. In the following, we will refer to the task of identifying the set of Arms that are Better than the Control in the presence of Subpopulations as the ABC-S problem.

Second, the practice of A/B/n testing often differs from a pure sequential experiment in that the experimenter cannot always fix a risk $\delta$ at the beginning and passively wait for the stopping time of the experiment without any time limitation. To address this issue, [11] proposed to define some notion of sequential "p-values" that can be monitored as the experiment progresses and used to terminate it. This notion was further used in the BAI setting in [18]. In this contribution, we elaborate on this idea by sequentially updating a suggested solution to the ABC-S problem *together with* a risk assessment for this suggestion. We show that, for any stopping time, the probability that the suggested solution is incorrect is indeed lower than the risk assessment. When the stopping time is selected as in usual fixed-confidence pure exploration, we recover the exact same guarantees but this view of the problem also provides useful results, for instance, if the experiment needs to be terminated prematurely.

**Related work.**   Pure exploration strategies have been studied in various settings: the identification of the best arm [8, 10], the identification of the top $m$ arms [3, 12, 9] identifying the arms that are better than a threshold [16, 4], or identifying all $\epsilon$-good arms [17]. As far as we know, this paper is the first to consider the problem of identifying all the arms better than a control. It is also the first to consider subpopulations in pure exploration tasks. While motivated by the example of online companies, we believe that the proposed algorithms are relevant to other domains where randomized controlled trials are used for learning. An example could be clinical trials: one may wish to identify all the alternative treatments that work better that some reference medical treatment. This would permit to choose among them taking into account different characteristics (some could be cheaper, using another molecule for avoiding allergy, etc.).

Close to the notion of the control is the notion of threshold. Locatelli et al. [16] propose an algorithm for identifying all arms above a given threshold. Their algorithm samples according to the significance

of a statistical test, and shares some similarities with the present article in the Gaussian case; however, the perspective is rather different: the authors consider the *fixed-budget setting*: the total number of samples is fixed, and the goal is then to minimize the probability of returning a wrong answer at the end. Here, the index of the control arm is known but its probability distribution is not.

In our work, the quality of the different arms is assessed with a weighted combination of its subpopulations means. Minimizing the estimation error of a convex combination of means through adaptive sampling was considered in [2] with the introduction of a *stratified estimator* that will naturally appear in our analysis.

The paper is organized as follows. In Section 2, we present the mathematical model and study the information-theoretic complexity of the problem, extending the lower bound of [10] to the ABC-S setting. We show how the complexity of the problem depends on the degree of interaction that one has with the subpopulations, introducing different modes of interaction to be defined in Figure 1 below. We also consider in detail the Gaussian case which gives rise to more interpretable results. Section 3 describes how to implement the proposed strategy, which involves the numerical resolution of non-trivial optimization problems. Finally, we provide the results of numerical experiments on synthetic and real data sets in Section 4.

## 2 The complexity of the ABC-S problem

### 2.1 Mathematical framework

A problem instance consists of the following ingredients. Known to the learner are the number of arms $K \geq 1$ in addition to the designated control arm 0, the number of subpopulations $J$ (a standard bandit being $J = 1$), and the vector $\boldsymbol{\beta} \in \mathbb{R}^J$ representing the relative importance of the subpopulations for the learning objective. We further make the stochastic assumption that samples from each arm $a$ (including the control) and subpopulation $i$ are drawn i.i.d. from an unknown probability distribution $\nu_{a,i}$ on $\mathbb{R}$, whose mean we will denote by $\mu_{a,i}$. The quality of arm $a$ is $\mu_a := \sum_{i=1}^{J} \beta_i \mu_{a,i}$ the combination of the means of the arms in the different populations. For $\boldsymbol{\beta} \in \mathbb{R}^J$ we define the ABC-S problem as the correct identification of the set

$$\mathcal{S}_{\boldsymbol{\beta}}(\boldsymbol{\mu}) := \left\{ a \in [K] \ \Big| \ \sum_{i=1}^{J} \beta_i \mu_{a,i} > \sum_{i=1}^{J} \beta_i \mu_{0,i} \right\} .$$

At every time step $t$, the algorithm selects an arm $A_t$ based on previous choices and outcomes and observes or selects (except when explicitly specified) the population type $I_t$. Upon the selection of the arm $A_t$ a reward $X_t$ is obtained. This defines a sigma-field generated by the observations up to time $t$ denoted $\mathcal{F}_t = \sigma(I_1, X_1, \ldots, I_t, X_t)$. The number of times arm $a$ was selected for subpopulation $i$ at time $t$ is denoted $N_{a,i}(t) := \sum_{s=1}^{t} \mathbb{1}(A_s = a, I_s = i)$ and the number of draws of arm $a$, $N_a(t) := \sum_{s=1}^{t} \mathbb{1}(A_s = a)$. We define the gap with the control arm and arm $a$, $\Delta_a := \mu_0 - \mu_a$.

**Modes of interaction** We consider four modes of interaction of the learner with the bandit, as specified in Figure 1 below. In any of the three passive modes of interaction (described in Figures 1b to 1d), we assume that the subpopulation $i$ represents a known proportion $\alpha_i$ of the total population, and hence that the sequence of subpopulations is drawn i.i.d. from the fixed and discrete distribution $I_t \sim \boldsymbol{\alpha} = (\alpha_1, \ldots, \alpha_J)$ with $\boldsymbol{\alpha} \in \Sigma_J := \{x \in [0,1]^J \mid \sum_i x_i = 1\}$ the $J$-dimensional simplex. Here $\boldsymbol{\alpha}$ is an exogenous parameter and can differ from $\boldsymbol{\beta}$ which is inherent to the learning objective and is also assumed to be known. Although it is most natural in many applications to consider that $\boldsymbol{\beta} = \boldsymbol{\alpha}$ (it is even necessary in the oblivious mode to make the estimation of the $\mu_a$'s feasible), Example 1 below describes a concrete scenario in which $\boldsymbol{\beta}$ has negative components.

The distributions $(\nu_{a,i})_{a,i}$ are assumed to belong the same one-parameter exponential family, $\mathcal{P} := \{(\nu_\theta)_\theta : d\nu_\theta/d\xi = \exp(\theta x - b(\theta))\}$, with $\xi$ a reference measure on $\mathbb{R}$ and $b : \Theta \subset \mathbb{R} \mapsto \mathbb{R}$. Every probability distribution $\nu_\theta$ in $\mathcal{P}$ is entirely defined by its mean $\dot{b}(\theta)$ [1]. We may hence identify any bandit instance with its matrix of means $\boldsymbol{\mu} \in \mathbb{R}^{(K+1) \times J}$ In addition, the Kullback-Leibler divergence between two distributions $\nu_\theta$ and $\nu_{\theta'} \in \mathcal{P}$ may be written in the following Bregman form:

$$d(\mu, \mu') = \mathrm{KL}(\nu_\theta, \nu_{\theta'}) = b(\theta') - b(\theta) - \dot{b}(\theta)(\theta' - \theta) ,$$

|  | 1. See $I_t \sim \boldsymbol{\alpha}$ | 1. Pick $A_t$ | 1. Pick $A_t$ |
|---|---|---|---|
| 1. Pick $A_t$ and $I_t$ | 2. Pick $A_t$ | 2. See $I_t \sim \boldsymbol{\alpha}$ | 2. Do *not* see $I_t \sim \boldsymbol{\alpha}$ |
| 2. See $X_t \sim \nu_{A_t,I_t}$ | 3. See $X_t \sim \nu_{A_t,I_t}$ | 3. See $X_t \sim \nu_{A_t,I_t}$ | 3. See $X_t \sim \nu_{A_t,I_t}$ |
| (a) *Active* mode | (b) *Proportional* mode | (c) *Agnostic* mode | (d) *Oblivious* mode. |

Figure 1: Modes of Interaction between Learner and Bandit in each round. In Active mode the learner determines the subpopulation, while in the right three passive modes it is sampled from $\boldsymbol{\alpha}$.

where $\mu = \dot{b}(\theta)$ and $\mu' = \dot{b}(\theta')$ correspond to the means of the two distributions $\nu_\theta$ and $\nu_{\theta'}$. We also use the notation $\mathrm{kl}(p, q)$ to denote the KL divergence of two Bernoulli distributions of parameter $p$ and $q$.

We define $\mathcal{L} := \{\boldsymbol{\mu} : \forall a \in [K] \cup \{0\}, \forall i \in [J], \nu_{a,i} \in \mathcal{P} \text{ and } \mu_0 \neq \mu_a\}$ the set of identifiable instances where no arm has the same weighted mean as the control. At every time step, the policies we consider output a risk assessment $\hat{\delta}_t$ together with a recommendation $\hat{\mathcal{S}}_t$. We focus on *safely calibrated* policies, that are defined as satisfying the following property

$$\forall \boldsymbol{\mu} \in \mathcal{L}, \ \forall \delta \in (0,1), \quad \mathbb{P}_{\boldsymbol{\mu}}\left(\exists t \geq 1 : \hat{\mathcal{S}}_t \neq \mathcal{S}_{\boldsymbol{\beta}}(\boldsymbol{\mu}) \ \cap \ \hat{\delta}_t \leq \delta\right) \leq \delta \ . \tag{1}$$

Finally, when fixing a level of risk $\delta$, we consider the stopping time associated to the filtration $\mathcal{F}_t$, $\tau_\delta = \inf\{t \geq 0, \ \hat{\delta}_t \leq \delta\}$. The objective is then to minimize the expected number of rounds necessary to obtain a level of risk of at most $\delta$. Contrary to usual $\delta$-PAC algorithms if stopped before $\tau_\delta$, the strategy still provides guarantees on the output set following Equation 1. In particular, safely calibrated policies have a sampling rule that does not depend on any pre-specified $\delta$, and as such they are $\delta$-PAC for any $\delta$.

**Example 1** ([Largest Profit Identification problem 13, p24]). *Consider a company choosing among $K$ product designs the model to mass produce. Each candidate design $k$ has an (equilibrium) sales price $\mu_{k,1}$ and production cost $\mu_{k,2}$. The goal is to find the model $k$ with the largest profit $\mu_{k,1} - \mu_{k,2}$. Prices and costs are currently unknown, but can be adaptively sampled. Sampling the "price" subpopulation $i = 1$ is typically implemented by performing user preference studies, taking questionnaires, etc. Samples from the "cost" subpopulation $i = 2$ involve rating manufacturing facilities, forecasting material and labor costs etc. This problem is interesting both in the BAI and ABC objectives. The importance vector is here $\boldsymbol{\beta} = (1, -1)$ and $\boldsymbol{\alpha}$ has to be set by the learner.*

## 2.2 General form of the sample complexity

Depending on the mode of interaction from Figure 1, the learner has a set of sampling constraints to satisfy, here denoted $\mathcal{C}$ and precisely defined in the next section. We define $\mathrm{Alt}(\boldsymbol{\mu})$, the different problem instances where the set of arms better than the control differs from that of the instance $\boldsymbol{\mu}$. Formally, $\mathrm{Alt}_{\boldsymbol{\beta}}(\boldsymbol{\mu}) := \{\boldsymbol{\lambda} \in \mathcal{L} \mid \mathcal{S}_{\boldsymbol{\beta}}(\boldsymbol{\lambda}) \neq \mathcal{S}_{\boldsymbol{\beta}}(\boldsymbol{\mu})\}$. This allows us to bound the sample complexity.

**Theorem 1.** *Let $\delta \in (0, 1)$ and $\boldsymbol{\beta} \in \mathbb{R}^J$. For any strategy satisfying Equation 1 and any $\boldsymbol{\mu} \in \mathcal{L}$, the expected number of rounds for the ABC-S problem for the agnostic, proportional and active mode satisfies:*

$$\mathbb{E}_{\boldsymbol{\mu}}[\tau_\delta] \geq T^\star(\boldsymbol{\mu}) \, \mathrm{kl}(\delta, 1 - \delta) \quad \text{and} \quad \liminf_{\delta \to 0} \frac{\mathbb{E}_{\boldsymbol{\mu}}[\tau_\delta]}{\ln(1/\delta)} \geq T^\star(\boldsymbol{\mu}) \ . \tag{2}$$

*where (recalling that $\lambda_a = \sum_{i=1}^J \beta_i \lambda_{a,i}$)*

$$T^\star(\boldsymbol{\mu})^{-1} = \sup_{\boldsymbol{w} \in \mathcal{C}} \inf_{\boldsymbol{\lambda} \in \mathrm{Alt}_{\boldsymbol{\beta}}(\boldsymbol{\mu})} \sum_{a=0}^K \sum_{i=1}^J w_{a,i} d(\mu_{a,i}, \lambda_{a,i}) \tag{3}$$

$$= \sup_{\boldsymbol{w} \in \mathcal{C}} \min_{b \neq 0} \inf_{\boldsymbol{\lambda} \in \mathcal{L}:\lambda_0=\lambda_b} \sum_{a \in \{0,b\}} \sum_{i=1}^J w_{a,i} d(\mu_{a,i}, \lambda_{a,i}) \ . \tag{4}$$

This result is established in Appendix A. $T^\star$ characterizes the difficulty of the learning problem.

## 2.3 Influence of the mode of interaction

We consider the four different modes governing the sampling rule as outlined in Figure 1. In the *agnostic* mode (Fig. 1c) an arm is first selected, after which the subpopulation type is observed. Mathematically, this brings the equality $\mathbb{E}_{\boldsymbol{\mu}}[N_{a,i}(T)] = \alpha_i \mathbb{E}_{\boldsymbol{\mu}}[N_a(T)]$ established in Lemma 2 and the independence constraint on the weights $\boldsymbol{w} \in \mathcal{C}_{\text{agnostic}} := \{w_{a,i} = \alpha_i u_a : (u_0, \ldots, u_K) \in \Sigma_{K+1}\}$.

In the *proportional* mode (Fig. 1b), $A_t$ is chosen based on $\mathcal{F}_{t-1}$ and the current subpopulation $I_t$. Here, the constraint is that the total number of pulls of the different arms in the subpopulation $i$ should respect the frequency of this subpopulation, i.e. $\sum_a \mathbb{E}_{\boldsymbol{\mu}}[N_{a,i}(T)] = \alpha_i T$. This induces a marginal constraint on the weights of the form $\boldsymbol{w} \in \mathcal{C}_{\text{prop}} := \{\boldsymbol{w} \in \Sigma_{(K+1)J} \mid \forall i \leq J, \sum_a w_{a,i} = \alpha_i\}$. This result is established in Lemma 3 reported in Appendix B.

In the *active* mode (Fig. 1a), the learner has an additional degree of freedom— she can ask for any subpopulation type at any round. In that case, $\boldsymbol{w} \in \mathcal{C}_{\text{active}} := \Sigma_{(K+1)J}$ is unconstrained.

By remarking that $\mathcal{C}_{\text{agnostic}} \subset \mathcal{C}_{\text{prop}} \subset \mathcal{C}_{\text{active}}$, and given the optimization program (3) solved to obtain the characteristic time, one immediately gets

$$\forall \boldsymbol{\mu} \in \mathcal{L}, \quad T_{\text{active}}^{\star}(\boldsymbol{\mu}) \leq T_{\text{proportional}}^{\star}(\boldsymbol{\mu}) \leq T_{\text{agnostic}}^{\star}(\boldsymbol{\mu}) . \tag{5}$$

Hence, as expected, the more control/information on the subpopulation the learner has, the faster she is able to identify the set of arms that are better than the control.

To compare with the *oblivious* mode, in which the subpopulation information is not even observed, we have to assume that $\boldsymbol{\alpha} = \boldsymbol{\beta}$. In that case, the arm rewards follow a mixture distribution: $X_t | A_t = a \sim \sum_{i=1}^{J} \alpha_i \nu_{a,i}$. In Proposition 4 reported in Appendix B.3, we properly define the characteristic time of an oblivious safely calibrated policy and prove that the joint convexity of Kullback-Leibler divergences implies that it is larger than its agnostic counterpart. This completes the picture of the ordering of the characteristic times by showing that, when $\boldsymbol{\alpha} = \boldsymbol{\beta}$,

$$\forall \boldsymbol{\mu} \in \mathcal{L}, \quad T_{\text{active}}^{\star}(\boldsymbol{\mu}) \leq T_{\text{proportional}}^{\star}(\boldsymbol{\mu}) \leq T_{\text{agnostic}}^{\star}(\boldsymbol{\mu}) \leq T_{\text{oblivious}}^{\star}(\boldsymbol{\mu}) . \tag{6}$$

Note that although we provide, in Section 3, algorithms to numerically compute the first three complexites, evaluating $T_{\text{oblivious}}^{\star}(\boldsymbol{\mu})$ would be much harder, as the mixture distributions can no more be parameterized by their mean only. Our current techniques do not yield a general-purpose practical algorithm that is asymptotically optimal in the *oblivious* mode for the ABC-S problem. In the Bernoulli case, however, as mixtures of Bernoulli distributions are Bernoulli distribution, one can use the single-population Bernoulli approach discussed in the next paragraph. For Gaussian distributions, one can use a suboptimal approach based on the observation that location mixtures of Gaussians with bounded means are sub-Gaussian (see Appendix B.3 for details).

## 2.4 Single population and relationship with best arm identification

In order to illustrate the nature of the the ABC-S problem, we make a detour through the single population case, that is, when $J = 1$. Given two weights $w_a, w_b$ and two means $\mu_a, \mu_b$, we introduce the minimum weighted transportation cost for moving the means to a common position.

$$d_{\text{mid}}(w_a, \mu_a, w_b, \mu_b) := \inf_v w_a d(\mu_a, v) + w_b d(\mu_b, v) = w_a d(\mu_a, v_{a,b}^{\star}) + w_b d(\mu_b, v_{a,b}^{\star})$$

where $v_{a,b}^{\star}$, the optimal common location, is the weighted average, i.e. $v_{a,b}^{\star} = \frac{w_a}{w_a + w_b} \mu_a + \frac{w_b}{w_a + w_b} \mu_b$.

**Constructing an instance in the alternative** When identifying all the arms better than a control, there are two different ways to obtain a close-by bandit model $\boldsymbol{\lambda}$ in the alternative. The first option consists in taking an arm which does not belong to $\mathcal{S}_{\boldsymbol{\beta}}(\boldsymbol{\mu})$ and to augment its mean on the alternative model such that it becomes above the control (or to reduce the mean of the control). Otherwise, it is possible to take an arm that is better than the control in the bandit model $\boldsymbol{\mu}$ and to shrink its mean such that it becomes lower than the control on the alternative (or augment the control). Note that the infimum over the alternative has the same expression in the two cases (see proof of Proposition 1 in Appendix A.2).

There is a priori no link between a BAI problem and an ABC one. In particular, in the BAI problem there are only $K + 1$ possible choices for the best arm while when looking for $\mathcal{S}_{\boldsymbol{\beta}}(\boldsymbol{\mu})$ there are up to

$2^K$ different sets to consider. Yet, the next proposition shows that the characteristic time $T^\star$ of any ABC problem with $J = 1$ subpopulation shares strong similarities with that of BAI problems.

**Proposition 1.** *Let $\delta \in (0,1)$ and $\boldsymbol{\mu} \in \mathcal{L}$. For any strategy satisfying Equation 1, Equation 2 holds with*

$$T^\star(\boldsymbol{\mu})^{-1} = \sup_{\boldsymbol{w} \in \Sigma_{K+1}} \inf_{\boldsymbol{\lambda} \in \mathrm{Alt}_\beta(\boldsymbol{\mu})} \sum_{a=0}^{K} w_a d(\mu_a, \lambda_a) = \sup_{\boldsymbol{w} \in \Sigma_{K+1}} \min_{b \neq 0} d_{\mathrm{mid}}(w_0, \mu_0, w_b, \mu_b) \ .$$

The proof is reported in Appendix A.2. Note that the expression of the sample complexity is really close to the one in the BAI setting (Garivier and Kaufmann [10, Lemma 3]) except that we consider all the indices different from the control here instead of the indices different from the best arm.

## 2.5 The Gaussian case

In this section, we consider the Gaussian case which is of interest as the characteristic time admits a more explicit expression, making it possible to further investigate the differences between the various modes of interaction. We will state our results for the heteroscedastic case, in particular to get a closed-form proxy for the Bernoulli case, where each variance is a function of the (unknown) mean.

**A/B testing** When $K = 1$ (one arm and the control arm), we are considering a standard A/B test with subpopulations and one can easily prove the following result (established in Appendix C).

**Proposition 2.** *For any $\boldsymbol{\mu} \in \mathcal{L}$ with $K = 1$ and $\nu_{a,i} = \mathcal{N}(\mu_{a,j}, \sigma_{a,j}^2)$ one has*

*1.* $T^\star_{\mathrm{agnostic}}(\boldsymbol{\mu}) = \dfrac{2\left(\sqrt{\sum_{i=1}^{J} \frac{\beta_i^2 \sigma_{0,i}^2}{\alpha_i}} + \sqrt{\sum_{i=1}^{J} \frac{\beta_i^2 \sigma_{1,i}^2}{\alpha_i}}\right)^2}{\Delta_1^2}$ *and* $w^\star_{a,i} = \dfrac{\alpha_i \sqrt{\sum_{i=1}^{J} \frac{\beta_i^2 \sigma_{a,i}^2}{\alpha_i}}}{\sqrt{\sum_{i=1}^{J} \frac{\beta_i^2 \sigma_{0,i}^2}{\alpha_i}} + \sqrt{\sum_{i=1}^{J} \frac{\beta_i^2 \sigma_{1,i}^2}{\alpha_i}}}$

*2.* $T^\star_{\mathrm{prop}}(\boldsymbol{\mu}) = \dfrac{2 \sum_{i=1}^{J} \frac{\beta_i^2}{\alpha_i} (\sigma_{0,i} + \sigma_{1,i})^2}{\Delta_1^2}$ *and* $\forall i \leq J, \forall a \in \{0,1\}, w^\star_{a,i} = \dfrac{\alpha_i \sigma_{a,i}}{\sigma_{0,i} + \sigma_{1,i}}$

*3.* $T^\star_{\mathrm{active}}(\boldsymbol{\mu}) = \dfrac{2\left(\sum_{i=1}^{J} |\beta_i|(\sigma_{0,i} + \sigma_{1,i})\right)^2}{\Delta_1^2}$ *and* $\forall i \leq J, \forall a \in \{0,1\}, w^\star_{a,i} = \dfrac{|\beta_i| \sigma_{a,i}}{\sum_{i=1}^{J} |\beta_i|(\sigma_{0,i} + \sigma_{1,i})}$

The optimal allocations in the *agnostic* and *proportional* cases are constrained by the proportion of the different subpopulations $\boldsymbol{\alpha}$, whereas, for the *active* mode, the optimal weights only depend on $\boldsymbol{\beta}$. In general, the optimal weights also depend on the subpopulation variances, as is well-known in stratified sampling estimation. Note however, that when (a) the subpopulations all have a common variance $\sigma^2$ and (b) $\boldsymbol{\beta} = \boldsymbol{\alpha}$, then the optimal allocations and the characteristic times are equal for the *agnostic*, the *proportional* and the *active* modes. In that case, $w^\star_{a,i} = \alpha_i/2$, which also corresponds to the well-known result in Gaussian A/B testing [14]. We have more generally observed that whenever the subpopulations have approximately the same variances, the *agnostic* and *proportional* modes yield very similar performances.

**Weight computation in the homoscedastic case** Even in scenarios where all subpopulation variances are equal to $\sigma^2$, the *active* mode remains very attractive in the cases where $\boldsymbol{\beta} \neq \boldsymbol{\alpha}$. The following proposition shows that in that case, the optimal weights for the ABC-S problem can be computed efficiently.

**Proposition 3** (Efficient computation in the Gaussian case). *With Gaussian distributions with a known variance $\sigma^2$, letting $(u_0^\star, \ldots, u_K^\star) = \mathrm{argmax}_{u \in \Sigma_{K+1}} \min_{b \neq 0} \frac{\Delta_b^2}{2\left(\frac{1}{u_0} + \frac{1}{u_b}\right)}$, the optimal weights for the active mode satisfy*

$$\forall a \in \{0, \ldots, K\}, \ \forall i \leq J, \ w^\star_{a,i} = u_a^\star \frac{|\beta_i|}{\sum_{i=1}^{J} |\beta_i|} \ .$$

*If, in addition $\boldsymbol{\alpha} = \boldsymbol{\beta}$, the above also holds for the agnostic and the proportional modes.*

The interesting part of Proposition 3 is that computing $(u_0^\star, \ldots, u_K^\star)$ can be done efficiently using Theorem 5 from [10]. The optimal weights of the ABC-S problem can be deduced from $u^\star$ without any further calculation.

## 3 Algorithms

To obtain our algorithms, we instantiate the Track-and-Stop algorithm template to our ABC-S problem. Garivier and Kaufmann [10] introduced Track-and-Stop and proved its asymptotic optimality in the BAI setting. Asymptotic optimality for general partition identification problems was subsequently established by Kaufmann and Koolen [13, Theorem 23] under the assumption of continuity of the oracle weights $\boldsymbol{\mu} \mapsto \boldsymbol{w}^*(\boldsymbol{\mu})$. Degenne and Koolen [6] show that the continuity assumption holds for all single-answer problems, in the upper-hemicontinuity sense, which they show implies asymptotic optimality of the Track-and-Stop (T-a-S) algorithm. These results directly apply to our ABC-S problem. Degenne et al. [7] interpret T-a-S as a noisy sequential equilibrium computation for the max-min problem from the lower bound (e.g. Equation 3) and develop computationally attractive variants including lazy iterative solution of the $\boldsymbol{w}^*$ problem, and optimistic gradients instead of forced exploration.

The details of our implementation are given in Appendix F. In short, we use the simple standard $\Theta(\sqrt{t})$ forced exploration rounds, a mode/subpopulation aware upgrade of the D-tracking scheme [10] (which is empirically superior to C-tracking) and we approximately and incrementally compute the oracle weights using the *AdaHedge vs Best Response* iterative saddle point solver from [7]. We use one single learner, instead of one per possible answer, as advocated in [7, Section 4]. Note that we are not affected by the non-convergence of D-tracking from [7, Appendix E], as our problem has a unique $\boldsymbol{w}^*$ because it is strictly concave in $\boldsymbol{w}$ (see Appendix E).

**The sampling rule**    The high level overview of the algorithm is as follows. We are given the number of arms $K$ and subpopulations $J$, the exponential family, the mode of interaction, the subpopulation importance coefficients $\boldsymbol{\beta}$ and, for passive modes, their natural frequencies $\boldsymbol{\alpha}$. The algorithm then proceeds in rounds $t = 1, 2, \ldots$ Each round $t$, it calculates the empirical frequencies $\hat{\boldsymbol{\mu}}_t \in \mathbb{R}^{(K+1) \times J}$ given by $\hat{\mu}_{a,i}(t) = \frac{1}{N_{a,i}(t)} \sum_{s=1}^{t} X_s \mathbf{1}\{A_s = a, I_s = i\}$. It then computes (a suitable approximation of) the maximiser (i.e. the oracle policy) $\boldsymbol{w}_t = \boldsymbol{w}^*(\hat{\boldsymbol{\mu}}_t) \in \Sigma_{(K+1) \times J}$ of problem (2). In the active mode, we "D-track" $\boldsymbol{w}_t$, i.e. we sample $(A_t, I_t) \in \operatorname{argmax}_{a,i} N_{a,i}(t-1) - t \boldsymbol{w}_t(a, i)$. In the proportional mode, the subpopulation $I_t$ is given and we "D-track" the conditional distribution of $\boldsymbol{w}_t$ on arms given the subpopulation, i.e. $A_t \in \operatorname{argmax}_a N_{a,I_t}(t-1) - t\alpha_{I_t} \boldsymbol{w}_t(a|I_t)$, where $\boldsymbol{w}_t(a, i) = \alpha_i \boldsymbol{w}_t(a|i)$. In the agnostic mode we "D-track" the marginal distribution of $\boldsymbol{w}_t$ on arms, i.e. $A_t \in \operatorname{argmax}_a N_a(t-1) - t\boldsymbol{w}_t(a)$. For each mode, this sampling strategy ensures that $N_{a,i}(t) \approx t\boldsymbol{w}_t(a, i) \approx t w^*_{a,i}(\boldsymbol{\mu})$, thus driving down the reported level of confidence as quickly as possible given the lower bound from Theorem 1.

**The recommendation**    Concluding each round, we recommend $\mathcal{S}_{\boldsymbol{\beta}}(\hat{\boldsymbol{\mu}}_t)$ at confidence level $\hat{\delta}(t) = \min\{\delta \in (0,1)|\Lambda(t) \geq \beta(t, \delta)\}$ obtained by inverting the threshold $\beta(t, \delta)$ at the GLR statistic

$$\Lambda(t) = \min_{b \neq 0} \inf_{\boldsymbol{\lambda} \in \mathcal{L}: \lambda_0 = \lambda_b} \sum_{a \in \{0, b\}} \sum_{i=1}^{J} N_{a,i}(t) d(\hat{\mu}_{a,i}(t), \lambda_{a,i}) . \tag{7}$$

**The threshold**    For the sharpest theoretically supported thresholds we refer to [13]. Namely, an ABC-S problem with $K$-arms and $J$-subpopulations has $2^K$ answers, and its *rank* [13, Definition 22] is $2J$, as can be read off from (4). By [13, Proposition 23] we have validity for $\beta(t, \delta) = 6J \ln \ln t + \ln \frac{1}{\delta} + K + 2J \cdot O(\ln \ln \frac{1}{\delta})$. In practice, we follow [10] and use instead the heavily stylized $\ln((1 + \ln t)/\delta)$ that omits several union bounds.

**Theorem 2.** *For every mode, Subpopulation Track-and-Stop is safely calibrated (Equation 1). Moreover, Subpopulation Track-and-Stop is asymptotically optimal and matches the lower bound from Theorem 1, in the sense that*

$$\text{for every bandit } \boldsymbol{\mu} \in \mathcal{L}, \ \lim_{\delta \to 0} \frac{\mathbb{E}[\tau_\delta]}{\ln(1/\delta)} = T^\star(\boldsymbol{\mu}) .$$

We include the proof in Appendix E.

# 4 Experiments

## 4.1 Simulations

We conduct numerical experiments to evaluate the proposed algorithms, focusing on Bernoulli bandit models, which are ubiquitous in practical applications.

In our experiments, in addition to our T-a-S algorithms with the various interaction modes, we include two more sampling rules for comparison: (1) uniform sampling as a baseline, and (2) the experimentally efficient *Best Challenger* heuristic inspired by [10], adapted to the ABC problem and denoted BC-ABC in the sequel. BC [10] for the BAI problem samples in every round the empirical best arm $\hat{a}_t$ or its best challenger, i.e. the arm $\hat{c}_t \neq \hat{a}_t$ at which the GLR statistic (Equation 7) reaches its minimum. Our BC-ABC adaptation samples in every round the control arm or the arm that yields the minimum GLR statistic $\Lambda(t)$, in the agnostic interaction mode (since $\Lambda(t)$ is subpopulation independent). For clearer comparison between the sampling strategies, all algorithms use the Chernoff stopping criterion [10] to determine either when to stop or output the risk assessment at a given time. We also opted for sampling rules independent from the confidence parameter $\delta$, because we are aiming for safely calibrated policies.

We first illustrate the fact that the T-a-S algorithm provides a correct –but rather conservative– assessment of the risk of its decision whatever the time it is stopped at. To do so, we generated 1000 bandit instances uniformly at random from [0, 1] with $K = 2$ arms. For each instance, we recorded the first time a certain risk assessment level is reached and the correctness of the algorithm's recommendation at that point. We map to each risk assessment level the proportion of errors across all instances. We chose two stopping rates that are not supported by theory but are recommended in practice [10]. Figure 2 (Left) illustrates the isotonic curve fitted on our observations and suggests that even the most lenient stopping threshold $\ln((\ln(t)+1)/\delta)$ results in much lower empirical probability of error than the risk assessment. In the following, we use the stopping threshold $\ln((\ln(t)+1)/\delta)$.

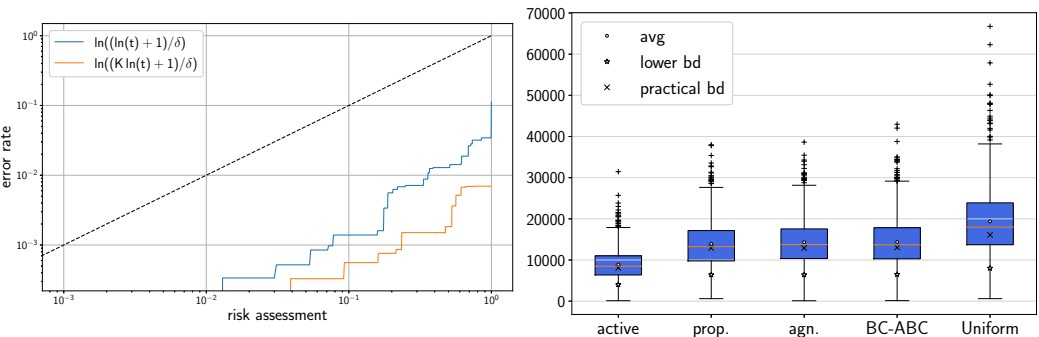

Figure 2: (Left) Risk assessment calibration on a log-log scale. (Right) Stopping time boxplot for $\boldsymbol{\mu} = [0.1\ 0.4\ 0.3; 0.2\ 0.5\ 0.2; 0.5\ 0.1\ 0.1] \in [0,1]^{(K+1)\times J}$ when $\boldsymbol{\beta} = [1/3, 1/3, 1/3], \boldsymbol{\alpha} = [0.4, 0.5, 0.1]$ with Bernoulli distributions.

In our second experiment [1], we generated 3000 Bernoulli bandit instances with $K = 2$ and a random number of subpopulations $J$ between 2 and 10. Each subpopulation-arm's mean $\mu_{a,i}$ is drawn uniformly at random from $[0, 1]$, and the subpopulation frequency vector $\boldsymbol{\alpha}$ is drawn from a Dirichlet(10) distribution. Table 1 reports the average stopping time of each algorithm across all bandit instances. On average, the T-a-S algorithms at all modes stop at similar times, and all adaptive sampling methods terminate faster than uniform sampling.

Table 1: Average stopping time. Description in text.

| T-a-S (active) | T-a-S (proportional) | T-a-S (agnostic) | BC-ABC | Uniform |
|---|---|---|---|---|
| 14871 | 15231 | 15444 | 15279 | 21586 |

---

[1]Code at `https://gitlab.com/ckatsimerou/abc_s_public`

To better understand the role of $\beta$ and $\alpha$, we ran the algorithms on a specific model (see Figure 2, Right) with $\alpha \neq \beta$. In this case, the optimal proportions are constrained by the frequencies of the subpopulation for passive interaction modes. The expected number of samples needed to identify the ABC-S solution is lower for the active policy, which has an additional degree of freedom in its sampling strategy. The *proportional* interaction mode and the *agnostic* interaction modes perform similarly. As expected, all the proposed strategies outperform the uniform sampling rule. We contrast the stopping time with the lower bound $\mathrm{kl}(\delta, 1-\delta)T^*(\boldsymbol{\mu})$, and with a more practical version, which indicates, approximately, the first time at which the GLR statistic crosses the threshold, i.e. solving $t = \ln((\ln(t) + 1)/\delta)T^*(\boldsymbol{\mu})$, as was done in [7]. All adaptive algorithms perform well on this instance, with their average runtime being very close to their respective practical bound.

## 4.2    Application to A/B/n experiment

We evaluate the algorithms on data collected from an actual A/B/n experiment, which compares different copies of a component of the webpage, in order to identify the ones better than the default copy. The metric of interest is whether the visitor clicked at least once during the experiment to the next page after getting exposed to one of the variants. For this setup we considered $K = 2$ copies competing against the control, with each copy being treated as an arm. Due to global traffic, the data exhibits strong seasonality patterns within a day, as seen in Figure 3a, in which every point corresponds to click-through rate per six hours (quarter of day) for 12 consecutive days. We treat the $J = 4$ seasons as i.i.d. subpopulations. Within each season we shuffled the data to eliminate the weekly trend.

The summary statistics of the dataset, together with the characteristic times and the optimal weights for each T-a-S mode can be found in Appendix G. Note that the small gaps between the arm means makes this practical ABC-S problem much harder than the synthetically generated examples.

We tested all algorithms described in Section 4.1. Each algorithm terminates when it reaches for the first time $\hat{\delta}_t \leq 0.1$ or outputs a risk assessment on the recommendation if it runs out of samples, which in this experiment occurs after $1.4 \cdot 10^7$ observations. Here, we weigh the importance $\beta$ of each season equally to its observed frequency $\alpha$. Doing so, we do not expect large performance discrepancies between the different T-a-S interaction modes, which is confirmed by their characteristic times (Appendix G). The observations from Fig. 3b are similar to the results from the numerical simulations: adaptive sampling achieves lower sample complexity over uniform sampling and T-a-S for the active interaction mode terminates faster than for the passive modes. All algorithms yield the correct recommendation, but not with the same risk assessment. All T-a-S algorithms terminated within the available sample size, BC-ABC almost terminated and output a risk assessment slightly above 0.1 and uniform's risk assessment was 0.67. Of course, when viewing seasonality as a subpopulation, the *active* mode is unrealistic, but it is still informative to see that it can be very economical in hard problems in which sampling the subpopulations actively is an option. In this instance, *proportional*, *agnostic* and *oblivious* modes terminated at similar times. However, we would recommend using the *proportional* mode, given that we expect it to never perform worse than the other passive modes on average. One should not be surprised by the curve for the uniform sampling, this policy was stopped before convergence because it ran out of samples.

Lastly, here we assumed that seasons occur in i.i.d. fashion, but in reality there is temporal dependence between them. This imposes extra constraints on the optimal weights and increases the sample complexity. However, we do not expect this to be detrimental for cases in which seasons alternate frequently and full cycles are observed often, as was the case with our example.

## 5    Conclusion

In this work, we considered the pure exploration task of identifying all the arms that are better than a control arm in the presence of subpopulations (ABC-S). We design asymptotically optimal policies for this problem under different assumptions on the mode of interaction between the learner and the bandit. We observed that the *active* mode, in which the learner decides which subpopulation it samples, may significantly reduce decision times. On the other hand, the other modes, in which the learner has to respect the natural proportions of the different subpopulations (i.e., in *proportional* and *agnostic* modes) produce more modest effects, except when the subpopulations differ significantly in

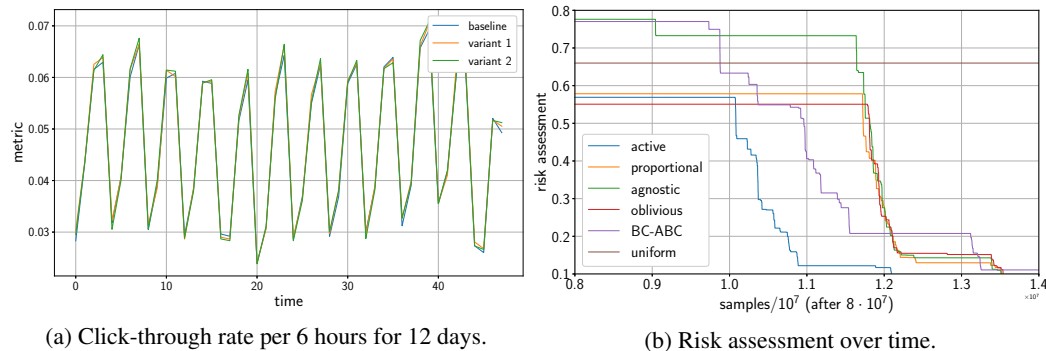

(a) Click-through rate per 6 hours for 12 days.

(b) Risk assessment over time.

Figure 3: Real data and results.

variances. Finally, we proposed a natural way to provide anytime decisions with risk guarantees in the Track-and-Stop framework.

## 6   Potential Societal Impact

The contributions presented in this work are mostly related to methods and, as such, do not have a direct expected societal impact. This being said, a potential concern that will need to be addressed more carefully in subsequent applications of these methods is the use of subpopulation information, which could be exploited to target specific user behaviour or characteristics. In the use case considered in Section 4.2, the subpopulations correspond to time slots that are used to model seasonality in the user responses, which does not raise any specific ethical concern. However, in cases where the subpopulations are formed using characteristics of individual users, the impact needs to be assessed more thoroughly. Note that in such cases, restricting to one of the more conservative modes of interaction (i.e. agnostic or even oblivious) may become necessary in order to prevent undue use of population-dependent information.

## Acknowledgment
The authors would like to thank the anonymous reviewers whose comments and questions helped improve the clarity of this manuscript. A. Garivier acknowledges the support of the Project IDEX-LYON of the University of Lyon, in the framework of the Programme Investissements d'Avenir (ANR-16-IDEX-0005), and Chaire SeqALO (ANR-20-CHIA-0020).

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
