^{\star}_{\text{active}}(\boldsymbol{\mu}) \leq T^{\star}_{\text{proportional}}(\boldsymbol{\mu}) \leq T^{\star}_{\text{agnostic}}(\boldsymbol{\mu}) . \tag{5}$$

Hence, as expected, the more control/information on the subpopulation the learner has, the faster she is able to identify the set of arms that are better than the control.

To compare with the *oblivious* mode, in which the subpopulation information is not even observed, we have to assume that $\boldsymbol{\alpha} = \boldsymbol{\beta}$. In that case, the arm rewards follow a mixture distribution: $X_t | A_t = a \sim \sum_{i=1}^J \alpha_i \nu_{a,i}$. In Proposition 4 reported in Appendix B.3, we properly define the characteristic time of an oblivious safely calibrated policy and prove that the joint convexity of Kullback-Leibler divergences implies that it is larger than its agnostic counterpart. This completes the picture of the ordering of the characteristic times by showing that, when $\boldsymbol{\alpha} = \boldsymbol{\beta}$,

$$\forall \boldsymbol{\mu} \in \mathcal{L}, \quad T^{\star}_{\text{active}}(\boldsymbol{\mu}) \leq T^{\star}_{\text{proportional}}(\boldsymbol{\mu}) \leq T^{\star}_{\text{agnostic}}(\boldsymbol{\mu}) \leq T^{\star}_{\text{oblivious}}(\boldsymbol{\mu}) . \tag{6}$$

Note that although we provide, in Section 3, algorithms to numerically compute the first three complexites, evaluating $T^{\star}_{\text{oblivious}}(\boldsymbol{\mu})$ would be much harder, as the mixture distributions can no more be parameterized by their mean only. Our current techniques do not yield a general-purpose practical algorithm that is asymptotically optimal in the *oblivious* mode for the ABC-S problem. In the Bernoulli case, however, as mixtures of Bernoulli distributions are Bernoulli distribution, one can use the single-population Bernoulli approach discussed in the next paragraph. For Gaussian distributions, one can use a suboptimal approach based on the observation that location mixtures of Gaussians with bounded means are sub-Gaussian (see Appendix B.3 for details).

## 2.4 Single population and relationship with best arm identification

In order to illustrate the nature of the the ABC-S problem, we make a detour through the single population case, that is, when $J = 1$. Given two weights $w_a, w_b$ and two means $\mu_a, \mu_b$, we introduce the minimum weighted transportation cost for moving the means to a common position.

$$d_{\text{mid}}(w_a, \mu_a, w_b, \mu_b) := \inf_v w_a d(\mu_a, v) + w_b d(\mu_b, v) = w_a d(\mu_a, v^{\star}_{a,b}) + w_b d(\mu_b, v^{\star}_{a,b})$$

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

*Proof.* Using the transportation lemma from [14] and recalling that $N_{a,i}(t)$ is the number of draws of arm $a$ in subpopulation $i$ up to time $t$, we have for any safely calibrated policies

$$\forall \boldsymbol{\lambda} \in \mathrm{Alt}_{\boldsymbol{\beta}}(\boldsymbol{\mu}), \ \sum_{i=1}^J \sum_{a=0}^K \mathbb{E}_{\boldsymbol{\mu}}[N_{a,i}(\tau_\delta)] d(\mu_{a,i}, \lambda_{a,i}) \geq \mathrm{kl}(\delta, 1-\delta) \ .$$

Therefore,

$$\mathrm{kl}(\delta, 1-\delta) \leq \inf_{\boldsymbol{\lambda} \in \mathrm{Alt}_{\boldsymbol{\beta}}(\boldsymbol{\mu})} \sum_{a=0}^K \sum_{i=1}^J \mathbb{E}_{\boldsymbol{\mu}}[N_{a,i}(\tau_\delta)] d(\mu_{a,i}, \lambda_{a,i})$$

$$= \mathbb{E}_{\boldsymbol{\mu}}[\tau_\delta] \inf_{\boldsymbol{\lambda} \in \mathrm{Alt}_{\boldsymbol{\beta}}(\boldsymbol{\mu})} \sum_{a=0}^K \sum_{i=1}^J \frac{\mathbb{E}_{\boldsymbol{\mu}}[N_{a,i}(\tau_\delta)]}{\mathbb{E}_{\boldsymbol{\mu}}[\tau_\delta]} d(\mu_{a,i}, \lambda_{a,i})$$

$$\leq \mathbb{E}_{\boldsymbol{\mu}}[\tau_\delta] \sup_{\boldsymbol{w} \in \mathcal{C}} \inf_{\boldsymbol{\lambda} \in \mathrm{Alt}_{\boldsymbol{\beta}}(\boldsymbol{\mu})} \sum_{a=0}^K \sum_{i=1}^J w_{a,i} d(\mu_{a,i}, \lambda_{a,i}) \ .$$

In the last inequality, we used the fact that the normalized expected numbers of draws satisfy the set of constraints defined by $\mathcal{C} \subset \Sigma_{(K+1)J}$. Using $\mathrm{kl}(\delta, 1-\delta) \sim \ln(1/\delta)$ when $\delta$ tends to 0 gives the first result.

We denote $\Lambda(\boldsymbol{w}, \boldsymbol{\lambda}, \boldsymbol{\mu}) := \sum_{a=0}^K \sum_{i=1}^J w_{a,i} d(\mu_{a,i}, \lambda_{a,i})$. To obtain the second result, we will simplify the expression of $T^\star(\boldsymbol{\mu})^{-1}$. Using that the KL divergences and the weights are positive, for $\boldsymbol{\lambda}$ to be in the alternative, one of the two following conditions need to be met: (1) there exists $a \in \mathcal{S}_{\boldsymbol{\beta}}(\boldsymbol{\mu})$ such that $\lambda_a < \lambda_0$. (2) there exists $a \in \mathcal{S}_{\boldsymbol{\beta}}^-(\boldsymbol{\mu}) := \{a \in [K] \mid \mu_a < \mu_0\}$ such that $\lambda_a > \lambda_0$.

For this reason, one has

$$\inf_{\boldsymbol{\lambda} \in \mathrm{Alt}_{\boldsymbol{\beta}}(\boldsymbol{\mu})} \Lambda(\boldsymbol{w}, \boldsymbol{\lambda}, \boldsymbol{\mu}) = \min \left( \min_{a \in \mathcal{S}_{\boldsymbol{\beta}}(\boldsymbol{\mu})} \inf_{\boldsymbol{\lambda}: \lambda_a < \lambda_0} \Lambda(\boldsymbol{w}, \boldsymbol{\lambda}, \boldsymbol{\mu}), \min_{a \in \mathcal{S}_{\boldsymbol{\beta}}^-(\boldsymbol{\mu})} \inf_{\boldsymbol{\lambda}: \lambda_a > \lambda_0} \Lambda(\boldsymbol{w}, \boldsymbol{\lambda}, \boldsymbol{\mu}) \right) \ .$$

We obtain the desired result by remarking that the inner optimization programs $\inf_{\boldsymbol{\lambda}}$ are each achieved on the boundary (the constraint being satisfied with equality) where they coincide, and that $\{1, \ldots, K\} = \mathcal{S}_{\boldsymbol{\beta}}(\boldsymbol{\mu}) \cup \mathcal{S}_{\boldsymbol{\beta}}^-(\boldsymbol{\mu})$. □

## A.2   Proof of Proposition 1

In the particular case when $J = 1$, the expression of the characteristic time can be simplified.

**Proposition 1.** *Let $\delta \in (0, 1)$ and $\boldsymbol{\mu} \in \mathcal{L}$. For any strategy satisfying Equation 1, Equation 2 holds with*

$$T^\star(\boldsymbol{\mu})^{-1} = \sup_{\boldsymbol{w} \in \Sigma_{K+1}} \inf_{\boldsymbol{\lambda} \in \mathrm{Alt}_\beta(\boldsymbol{\mu})} \sum_{a=0}^{K} w_a d(\mu_a, \lambda_a) = \sup_{\boldsymbol{w} \in \Sigma_{K+1}} \min_{b \neq 0} d_{\mathrm{mid}}(w_0, \mu_0, w_b, \mu_b) \ .$$

*Proof.* The first part of the proof can be obtained using similar argument than for Theorem 1. The missing part is the simplification of the expression of $T^\star(\boldsymbol{\mu})$.

We denote $\Lambda(\boldsymbol{w}, \boldsymbol{\lambda}, \boldsymbol{\mu}) := \sum_{a=0}^{K} w_a d(\mu_a, \lambda_a)$. Following the reasoning from the proof of Theorem 1 one of the two following conditions needs to be met: (1) there exists $a \in \mathcal{S}_\beta(\boldsymbol{\mu})$ such that $\lambda_a < \lambda_0$. (2) there exists $a \in \mathcal{S}_{\boldsymbol{\beta}}^-(\boldsymbol{\mu}) := \{a \in [K] \mid \mu_a < \mu_0\}$ such that $\lambda_a > \lambda_0$.

For this reason, one has

$$\inf_{\boldsymbol{\lambda} \in \mathrm{Alt}_\beta(\boldsymbol{\mu})} \Lambda(\boldsymbol{w}, \boldsymbol{\lambda}, \boldsymbol{\mu}) = \min \left( \min_{a \in \mathcal{S}_\beta(\boldsymbol{\mu})} \inf_{\boldsymbol{\lambda}:\lambda_a < \lambda_0} \Lambda(\boldsymbol{w}, \boldsymbol{\lambda}, \boldsymbol{\mu}), \min_{a \in \mathcal{S}_{\boldsymbol{\beta}}^-(\boldsymbol{\mu})} \inf_{\boldsymbol{\lambda}:\lambda_a > \lambda_0} \Lambda(\boldsymbol{w}, \boldsymbol{\lambda}, \boldsymbol{\mu}) \right) \ .$$

In this simpler case, it is possible to obtain an explicit formula for this infimum. We start from

$$T^\star(\boldsymbol{\mu})^{-1} = \sup_{\boldsymbol{w} \in \Sigma_{K+1}} \min \left( \min_{a \in \mathcal{S}_\beta(\boldsymbol{\mu})} \inf_{\boldsymbol{\lambda}:\lambda_a < \lambda_0} \Lambda(\boldsymbol{w}, \boldsymbol{\lambda}, \boldsymbol{\mu}), \min_{a \in \mathcal{S}_{\boldsymbol{\beta}}^-(\boldsymbol{\mu})} \inf_{\boldsymbol{\lambda}:\lambda_a > \lambda_0} \Lambda(\boldsymbol{w}, \boldsymbol{\lambda}, \boldsymbol{\mu}) \right) \ .$$

Let us focus on the case, $\lambda_a < \lambda_0$ and fix an index $a \in \mathcal{S}_\beta(\boldsymbol{\mu})$. $\Lambda$ is always smaller when all the $\lambda_b$ for $b \neq 0$ and $b \neq a$ coincides with $\mu_b$. This gives,

$$\min_{a \in \mathcal{S}_\beta(\boldsymbol{\mu})} \inf_{\boldsymbol{\lambda}:\lambda_a < \lambda_0} \Lambda(\boldsymbol{w}, \boldsymbol{\lambda}, \boldsymbol{\mu}) = \min_{a \in \mathcal{S}_\beta(\boldsymbol{\mu})} \inf_{\boldsymbol{\lambda}:\lambda_a \leq \lambda_0} w_0 d(\mu_0, \lambda_0) + w_a d(\mu_a, \lambda_a) \ .$$

We consider the Lagrangian function, $L(\lambda_0, \lambda_a, q) = w_0 d(\mu_0, \lambda_0) + w_a d(\mu_a, \lambda_a) + q(\lambda_a - \lambda_0)$. Differentiating with respect to $\lambda_0$ and $\lambda_a$ brings the condition

$$\lambda_0^\star = \lambda_a^\star = \lambda_{a,0}^\star = \mathrm{argmin}_\lambda w_0 d(\mu_0, \lambda) + w_a d(\mu_a, \lambda) = \frac{w_0}{w_0 + w_a} \mu_0 + \frac{w_a}{w_0 + w_a} \mu_a \ .$$

Recalling, $d_{\mathrm{mid}}(w_a, \mu_a, w_b, \mu_b) := \inf_v w_a d(\mu_a, v) + w_b d(\mu_b, v)$ one has,

$$\min_{a \in \mathcal{S}_\beta(\boldsymbol{\mu})} \inf_{\boldsymbol{\lambda}:\lambda_a < \lambda_0} \Lambda(\boldsymbol{w}, \boldsymbol{\lambda}, \boldsymbol{\mu}) = \min_{a \in \mathcal{S}_\beta(\boldsymbol{\mu})} d_{\mathrm{mid}}(w_0, \mu_0, w_a, \mu_a) \ . \tag{8}$$

Solving the optimization program for $a \in \mathcal{S}_{\boldsymbol{\beta}}^-(\boldsymbol{\mu})$ and under the constraint $\lambda_a > \lambda_0$, gives the exact same set of constraints and optimal solution, i.e.

$$\min_{a \in \mathcal{S}_{\boldsymbol{\beta}}^-(\boldsymbol{\mu})} \inf_{\boldsymbol{\lambda}:\lambda_a > \lambda_0} \Lambda(\boldsymbol{w}, \boldsymbol{\lambda}, \boldsymbol{\mu}) = \min_{a \in \mathcal{S}_{\boldsymbol{\beta}}^-(\boldsymbol{\mu})} d_{\mathrm{mid}}(w_0, \mu_0, w_a, \mu_a) \ . \tag{9}$$

Bringing Equation 8 and Equation 9 together and remarking that $[K] = \mathcal{S}_\beta(\boldsymbol{\mu}) \cup \mathcal{S}_{\boldsymbol{\beta}}^-(\boldsymbol{\mu})$ gives the announced result. $\square$

### A.3 Link between ABC and BAI

In the particular case, of Gaussian distributions with a known variance $\sigma^2$, for any $\boldsymbol{\mu} \in \mathcal{L}$, one can easily create a BAI instance with the same characteristic time as $T^\star(\boldsymbol{\mu})$.

**Lemma 1.** *Let $\boldsymbol{\mu} \in \mathcal{L}$ where all the arms are Gaussian distributions with known variance $\sigma^2$. Let $\mu_0$ be the mean of the control arm. We define $\widetilde{\boldsymbol{\mu}}$ as follows*

$$\widetilde{\mu}_k = \begin{cases} 2\mu_0 - \mu_k & \text{if } \mu_k > \mu_0 \\ \mu_k & \text{otherwise.} \end{cases}$$

*By denoting $T_{BAI}^\star(\boldsymbol{\mu})$ the characteristic time for the BAI problem with a bandit instance $\boldsymbol{\mu}$, then one has*

$$T^\star(\boldsymbol{\mu}) = T_{BAI}^\star(\widetilde{\boldsymbol{\mu}}) \ .$$

*Proof.* In the particular case of Gaussian distributions with known variance $\sigma^2$, easy calculation brings

$$T^\star(\boldsymbol{\mu})^{-1} = \sup_{w\in\Sigma_{K+1}} \min_{b\neq 0} d_{\mathrm{mid}}(w_0,\mu_0,w_b,\mu_b) = \min_{b\neq 0} \frac{(\mu_0-\mu_b)^2}{2}((1-\alpha_b)^2 w_0 + \alpha_b^2 w_b)$$

with $\alpha_b = \frac{w_0}{w_b+w_0}$.

On the instance $\tilde{\boldsymbol{\mu}}$, first note that $\tilde{\mu}_0$ is the best arm by construction. For an index $k$ such that $\mu_k > \mu_0$, one has

$$\tilde{\mu}_0 - \tilde{\mu}_k = \mu_0 - (2\mu_0 - \mu_k) = \mu_k - \mu_0 > 0 \ .$$

For this reason, $\forall k \in [K], \tilde{\mu}_0 > \tilde{\mu}_k$. Using Lemma 3 from [10], by defining $I_\gamma(\mu_1,\mu_2) := \gamma d(\mu_1, \gamma\mu_1 + (1-\gamma)\mu_2) + (1-\gamma)d(\mu_2, \gamma\mu_1 + (1-\gamma)\mu_2)$, one has

$$T^\star_{\mathrm{BAI}}(\widetilde{\boldsymbol{\mu}}) = \sup_{w\in\Sigma_{K+1}} \min_{b\neq 0} (w_0+w_b)I_{\frac{w_0}{w_0+w_b}}(\mu_0,\mu_b) \ .$$

Furthermore, letting $\alpha_b = w_0/(w_0+w_b)$:

$$(w_0+w_b)I_{\alpha_b}(\mu_0,\mu_b) = (w_0+w_b)\left(\alpha_b\frac{(1-\alpha_b)^2(\mu_0-\mu_b)^2}{2\sigma^2} + (1-\alpha_b)\frac{\alpha_b^2(\mu_0-\mu_b)^2}{2\sigma^2}\right)$$

$$= \frac{(\mu_0-\mu_b)^2}{2\sigma^2}\left((1-\alpha_b)^2 w_0 + \alpha_b^2 w_b\right) \ .$$

Plugging this in the expression of $T^\star_{\mathrm{BAI}}(\widetilde{\boldsymbol{\mu}})$ gives the announced result. $\qquad\square$

# B  Results for specific modes of interaction

## B.1  Agnostic mode

**Lemma 2.** *For any agnostic policy where $A_t$ is chosen knowing $\mathcal{F}_{t-1}$ but independently from $I_t$, when defining $N_{a,j}(t) = \sum_{s=1}^t \mathbb{1}(A_s = a \cap I_s = j)$ and $N_a(t) = \sum_{s=1}^t \mathbb{1}(A_s = a)$, then*

$$\forall a \in \{0,\ldots,K\}, \forall j \in \{1,\ldots,J\}, \forall t \geq 1, \quad \mathbb{E}_{\boldsymbol{\mu}}[N_{a,j}(t)] = \alpha_j \mathbb{E}_{\boldsymbol{\mu}}[N_a(t)]$$

*Proof.*

$$\mathbb{E}_{\boldsymbol{\mu}}[N_{a,j}(t)] = \sum_{s=1}^t \mathbb{P}(A_s = a \cap I_s = j) = \sum_{s=1}^t \mathbb{P}_{\boldsymbol{\mu}}(A_s = a|I_s = j)\mathbb{P}(I_s = j)$$

$$= \sum_{s=1}^t \alpha_j \mathbb{P}_{\boldsymbol{\mu}}(A_s = a|I_s = j) = \sum_{s=1}^t \alpha_j \mathbb{P}_{\boldsymbol{\mu}}(A_s = a)$$

$$= \alpha_j \mathbb{E}_{\boldsymbol{\mu}}[N_a(t)] \ ,$$

where in the third equality, we have used that the action $A_t$ is selected independently from the population indicator $I_t$. $\qquad\square$

## B.2  Proportional mode

**Lemma 3.** *For any proportional policy where $A_t$ is chosen knowing $\mathcal{F}_{t-1}$ and $I_t$, when defining $N_{a,j}(t) = \sum_{s=1}^t \mathbb{1}(A_s = a \cap I_s = j)$ and $N_a(t) = \sum_{s=1}^t \mathbb{1}(A_s = a)$, then*

$$\forall j \in \{1,\ldots,J\}, \forall t \geq 1, \quad \sum_{a=0}^K \mathbb{E}_{\boldsymbol{\mu}}[N_{a,j}(t)] = \alpha_j t \ .$$

*Proof.*

$$\sum_{a=0}^K \mathbb{E}_{\boldsymbol{\mu}}[N_{a,j}(t)] = \sum_{s=1}^t\sum_{a=0}^K \mathbb{E}_{\boldsymbol{\mu}}\left[\mathbb{1}(I_s = j)\mathbb{1}(A_s = a)\right] = \sum_{s=1}^t \mathbb{E}_{\boldsymbol{\mu}}\left[\mathbb{1}(I_s = j)\sum_{a=0}^K \mathbb{1}(A_s = a)\right]$$

$$= \sum_{s=1}^t \mathbb{P}_{\boldsymbol{\mu}}(I_s = j) = \alpha_j t \ .$$

$$\square$$

## B.3 Oblivious mode

In the oblivious mode, the subpopulations can not be observed by the learner. In this case, we have

1. $\mathbb{E}\left[X_t | A_t = a\right] = \sum_{i=1}^{J} \alpha_i \mu_{a,i}$ ,
2. $X_t | A_t = a \sim \sum_{i=1}^{J} \alpha_i \nu_{a,i}$ .

While with observable subpopulations the distributions are entirely characterized by their means, this is no longer the case with mixture distributions. In particular, this requires defining a different alternative.

$$\mathrm{Alt}(\nu) := \{\nu' : \forall a, \nu'_a = \sum_{i=1}^{J} \alpha_i \nu'_{a,i} \text{ with } \nu'_{a,i} \in \mathcal{P} \text{ and } \mathcal{S}(\nu') \neq \mathcal{S}(\nu) \} .$$

**Proposition 4.** *Let $\delta \in (0,1)$ and $\boldsymbol{\beta} \in \mathbb{R}^J$. For any oblivious strategy satisfying Equation 1 and any $\boldsymbol{\mu} \in \mathcal{L}$, the characteristic time satisfies*

$$\mathbb{E}_{\boldsymbol{\mu}}[\tau_\delta] \geq T^\star_{\mathrm{oblivious}}(\boldsymbol{\mu}) \, \mathrm{kl}(\delta, 1-\delta) \quad \text{and} \quad \liminf_{\delta \to 0} \frac{\mathbb{E}_{\boldsymbol{\mu}}[\tau_\delta]}{\ln(1/\delta)} \geq T^\star_{\mathrm{oblivious}}(\boldsymbol{\mu}) .$$

*where*

$$T^\star_{\mathrm{oblivious}}(\boldsymbol{\mu})^{-1} = \sup_{\boldsymbol{w} \in \Sigma_{K+1}} \inf_{\nu' \in \mathrm{Alt}(\nu)} \sum_{a=0}^{K} w_a \mathrm{KL}\left(\sum_{i=1}^{J} \alpha_i \nu_{a,i}, \sum_{i=1}^{J} \alpha_i \nu'_{a,i}\right) . \tag{10}$$

*Furthermore,*

$$\forall \boldsymbol{\mu} \in \mathcal{L}, \quad T^\star_{\mathrm{oblivious}}(\boldsymbol{\mu}) \geq T^\star_{\mathrm{agnostic}}(\boldsymbol{\mu}) .$$

*Proof.* Using the transportation lemma from [14] we have for any safely calibrated oblivious policy

$$\forall \nu' \in \mathrm{Alt}(\nu), \, \sum_{a=0}^{K} \mathbb{E}_{\boldsymbol{\mu}}[N_a(\tau_\delta)] \mathrm{KL}\left(\sum_{i=1}^{J} \alpha_i \nu_{a,i}, \sum_{i=1}^{J} \alpha_i \nu'_{a,i}\right) \geq \mathrm{kl}(\delta, 1-\delta) .$$

Therefore,

$$\mathrm{kl}(\delta, 1-\delta) \leq \inf_{\nu' \in \mathrm{Alt}(\nu)} \sum_{a=0}^{K} \mathbb{E}_{\boldsymbol{\mu}}[N_a(\tau_\delta)] \mathrm{KL}\left(\sum_{i=1}^{J} \alpha_i \nu_{a,i}, \sum_{i=1}^{J} \alpha_i \nu'_{a,i}\right)$$

$$= \mathbb{E}_{\boldsymbol{\mu}}[\tau_\delta] \inf_{\nu' \in \mathrm{Alt}(\nu)} \sum_{a=0}^{K} \frac{\mathbb{E}_{\boldsymbol{\mu}}[N_a(\tau_\delta)]}{\mathbb{E}_{\boldsymbol{\mu}}[\tau_\delta]} \mathrm{KL}\left(\sum_{i=1}^{J} \alpha_i \nu_{a,i}, \sum_{i=1}^{J} \alpha_i \nu'_{a,i}\right)$$

$$\leq \mathbb{E}_{\boldsymbol{\mu}}[\tau_\delta] \sup_{\boldsymbol{w} \in \Sigma_{K+1}} \inf_{\nu' \in \mathrm{Alt}(\nu)} \sum_{a=0}^{K} w_a \mathrm{KL}\left(\sum_{i=1}^{J} \alpha_i \nu_{a,i}, \sum_{i=1}^{J} \alpha_i \nu'_{a,i}\right) .$$

Using $\mathrm{kl}(\delta, 1-\delta) \sim \ln(1/\delta)$ when $\delta$ tends to 0 gives the first result.

Using the joint convexity of the KL divergence one gets

$$\mathrm{KL}\left(\sum_{i=1}^{J} \alpha_i \nu_{a,i}, \sum_{i=1}^{J} \alpha_i \nu'_{a,i}\right) \leq \sum_{i=1}^{J} \alpha_i \mathrm{KL}(\nu_{a,i}, \nu'_{a,i}) .$$

Assuming that the mean of $\nu'_{a,i} = \lambda_{a,i}$ and recalling that for distributions in $\mathcal{P}$, one has $\mathrm{KL}(\nu_{a,i}, \nu'_{a,i}) = d(\mu_{a,i}, \lambda_{a,i})$, we deduce,

$$T^\star_{\mathrm{oblivious}}(\boldsymbol{\mu})^{-1} = \sup_{w \in \Sigma_{K+1}} \inf_{\nu' \in \mathrm{Alt}(\nu)} \sum_{a=0}^{K} w_a \mathrm{KL}\left(\sum_{i=1}^{J} \alpha_i \nu_{a,i}, \sum_{i=1}^{J} \alpha_i \nu'_{a,i}\right)$$

$$\leq \sup_{w \in \Sigma_{K+1}} \inf_{\boldsymbol{\lambda} \in \mathrm{Alt}(\boldsymbol{\mu})} \sum_{a=0}^{K} \sum_{i=1}^{J} \alpha_i w_a d(\mu_{a,i}, \lambda_{a,i})$$

$$= T^\star_{\mathrm{agnostic}}(\boldsymbol{\mu})^{-1} .$$

$\square$

Except in the case of Bernoulli distributions –where the mixture is also a Bernoulli distribution–, finding a strategy that matches $T^\star_{\text{oblivious}}(\boldsymbol{\mu})^{-1}$ is a hard task. However, one may use the following lemma to treat the mixture in a sub-optimal way, based on the fact that it exhibits sub-gaussian behavior.

**Lemma 4** (Sub-gaussianity of mixture). *For each $\mu \in \mathbb{R}$, assume that $\nu_\mu$ is a distribution on $\mathbb{R}$ with mean $\mathbb{E}_{X \sim \nu_\mu}[X] = \mu$ that is $\sigma^2$-sub-Gaussian, meaning that $\mathbb{E}_{X \sim \nu_\mu}\left[e^{\lambda(X-\mu)}\right] \leq e^{\sigma^2 \lambda^2/2}$ for any $\lambda \in \mathbb{R}$. Further let $\alpha(\mu)$ be a prior on $\mu$ with mean $m$ that is itself $\eta^2$-sub-Gaussian, meaning that $\mathbb{E}_{\mu \sim \alpha}\left[e^{\lambda(\mu-m)}\right] \leq e^{\lambda^2 \eta^2/2}$. Then the mixture distribution $Q = \mathbb{E}_{\mu \sim \alpha}[\nu_\mu]$ is $\sigma^2 + \eta^2$ sub-Gaussian.*

*Proof.* The mixture distribution obviously has mean $\mathbb{E}_{X \sim Q}[X] = m$ and

$$
\begin{aligned}
\mathbb{E}_{X \sim Q}\left[e^{\lambda(X-m)}\right] &= \mathbb{E}_{\mu \sim \alpha}\left[e^{\lambda(\mu-m)} \mathbb{E}_{X \sim \nu_\mu}\left[e^{\lambda(X-\mu)}\right]\right] \\
&\leq \mathbb{E}_{\mu \sim \alpha}\left[e^{\lambda(\mu-m)}\right] e^{\sigma^2 \lambda^2/2} \\
&\leq e^{(\sigma^2 + \eta^2)\lambda^2/2} .
\end{aligned}
$$

$\square$

In particular, if $\alpha$ is supported on $[\pm M]$, then $\alpha$ is $M^2$ sub-Gaussian, and hence $Q$ is $(\sigma^2 + M^2)$ sub-Gaussian.

## C Optimal allocations in the Gaussian case for $K = 1$ (A/B testing)

**Lemma 5.** *When $K = 1$ with Gaussian distributions such that $\nu_{a,i} = \mathcal{N}(\mu_{a,i}, \sigma_{a,i}^2)$ the following holds*

$$
\inf_{\boldsymbol{\lambda}:\lambda_0 = \lambda_1} \sum_{i=1}^J w_{0,i} d(\mu_{0,i}, \lambda_{0,i}) + \sum_{i=1}^J w_{1,i} d(\mu_{1,i}, \lambda_{1,i}) = \frac{\Delta_1^2}{2 \sum_{i=1}^J \beta_i^2 \left(\frac{\sigma_{0,i}^2}{w_{0,i}} + \frac{\sigma_{1,i}^2}{w_{1,i}}\right)} .
$$

*Proof.* One has for $b \in \{0, 1\}$,

$$
d(\mu_{b,i}, \lambda_{b,i}) = \frac{(\lambda_{b,i} - \mu_{b,i})^2}{2\sigma_{b,i}^2} .
$$

Using the result from Theorem 1 for the case $K = 1$, the following holds

$$
\inf_{\boldsymbol{\lambda} \in \text{Alt}_\beta(\boldsymbol{\mu})} \sum_{a=0}^1 \sum_{i=1}^J w_{a,i} d(\mu_{a,i}, \lambda_{a,i}) = \min_{\boldsymbol{\lambda} \in \mathcal{L}:\lambda_0 = \lambda_1} \sum_{a=0}^1 \sum_{i=1}^J w_{a,i} d(\mu_{a,i}, \lambda_{a,i}) .
$$

We introduce

$$
L(\lambda_0, \lambda_1, q) = \sum_{i=1}^J w_{0,i} \frac{(\lambda_{0,i} - \mu_{0,i})^2}{2\sigma_{0,i}^2} + \sum_{i=1}^J w_{1,i} \frac{(\lambda_{1,i} - \mu_{1,i})^2}{2\sigma_{1,i}^2} + q\left(\sum_{i=1}^J \beta_i(\lambda_{0,i} - \lambda_{1,i})\right) .
$$

One has,

$$
\min_{\boldsymbol{\lambda} \in \mathcal{L}:\lambda_0 = \lambda_1} \sum_{a=0}^1 \sum_{i=1}^J w_{a,i} d(\mu_{a,i}, \lambda_{a,i}) = \sup_{q \in \mathbb{R}} \inf_{\boldsymbol{\lambda} \in \mathcal{L}} L(\lambda_0, \lambda_1, q) .
$$

Differentiating with respect to $\lambda_{0,i}$ and $\lambda_{1,i}$ brings the conditions

$$
\lambda_{0,i} = \mu_{0,i} - \frac{q \beta_i \sigma_{0,i}^2}{w_{0,i}} \quad \text{and} \quad \lambda_{1,i} = \mu_{1,i} + \frac{q \beta_i \sigma_{1,i}^2}{w_{1,i}} .
$$

Plugging these values back in $L$ gives the function

$$
f(q) = -\frac{q^2}{2} \sum_{i=1}^J \beta_i^2 \left(\frac{\sigma_{0,i}^2}{w_{0,i}} + \frac{\sigma_{1,i}^2}{w_{1,i}}\right) + q \sum_{i=1}^J \beta_i(\mu_{0,i} - \mu_{1,i}) .
$$

Easy calculations show that the maximum of the function $f$ is attained for

$$q^\star = \frac{\sum_{i=1}^{J} \beta_i(\mu_{0,i} - \mu_{1,i})}{\sum_{i=1}^{J} \beta_i^2 \left( \frac{\sigma_{0,i}^2}{w_{0,i}} + \frac{\sigma_{1,i}^2}{w_{1,i}} \right)} .$$

Plugging this value back in the expression of $f$,

$$f(q^\star) = \frac{\Delta_1^2}{2 \sum_{i=1}^{J} \beta_i^2 \left( \frac{\sigma_{0,i}^2}{w_{0,i}} + \frac{\sigma_{1,i}^2}{w_{1,i}} \right)} .$$

$\square$

**Proposition 2.** *For any $\boldsymbol{\mu} \in \mathcal{L}$ with $K = 1$ and $\nu_{a,i} = \mathcal{N}(\mu_{a,j}, \sigma_{a,j}^2)$ one has*

1. $T_{\text{agnostic}}^\star(\boldsymbol{\mu}) = \frac{2\left( \sqrt{\sum_{i=1}^{J} \frac{\beta_i^2 \sigma_{0,i}^2}{\alpha_i}} + \sqrt{\sum_{i=1}^{J} \frac{\beta_i^2 \sigma_{1,i}^2}{\alpha_i}} \right)^2}{\Delta_1^2}$ and $w_{a,i}^\star = \frac{\alpha_i \sqrt{\sum_{i=1}^{J} \frac{\beta_i^2 \sigma_{a,i}^2}{\alpha_i}}}{\sqrt{\sum_{i=1}^{J} \frac{\beta_i^2 \sigma_{0,i}^2}{\alpha_i}} + \sqrt{\sum_{i=1}^{J} \frac{\beta_i^2 \sigma_{1,i}^2}{\alpha_i}}}$

2. $T_{\text{prop}}^\star(\boldsymbol{\mu}) = \frac{2 \sum_{i=1}^{J} \frac{\beta_i^2}{\alpha_i}(\sigma_{0,i} + \sigma_{1,i})^2}{\Delta_1^2}$ and $\forall i \leq J, \forall a \in \{0, 1\}, \; w_{a,i}^\star = \frac{\alpha_i \sigma_{a,i}}{\sigma_{0,i} + \sigma_{1,i}}$

3. $T_{\text{active}}^\star(\boldsymbol{\mu}) = \frac{2\left( \sum_{i=1}^{J} |\beta_i|(\sigma_{0,i} + \sigma_{1,i}) \right)^2}{\Delta_1^2}$ and $\forall i \leq J, \forall a \in \{0, 1\}, \; w_{a,i}^\star = \frac{|\beta_i| \sigma_{a,i}}{\sum_{i=1}^{J} |\beta_i|(\sigma_{0,i} + \sigma_{1,i})}$

*Proof.*

**Agnostic mode** From the Lemma 2, we have

$$T_{\text{agnostic}}^\star(\boldsymbol{\mu})^{-1} = \sup_{\boldsymbol{w} \in \mathcal{C}_{\text{agnostic}}} \inf_{\boldsymbol{\lambda} \in \text{Alt}_{\boldsymbol{\beta}}(\boldsymbol{\mu})} \sum_{a=0}^{1} \sum_{i=1}^{J} w_{a,i} d(\mu_{a,i}, \lambda_{a,i})$$

$$= \sup_{\boldsymbol{w} \in \mathcal{C}_{\text{agnostic}}} \inf_{\boldsymbol{\lambda}: \lambda_0 = \lambda_1} \sum_{a=0}^{1} \sum_{i=1}^{J} w_{a,i} d(\mu_{a,i}, \lambda_{a,i}) \quad \text{(Theorem 1)}$$

$$= \sup_{\boldsymbol{w} \in \mathcal{C}_{\text{agnostic}}} \frac{\Delta_1^2}{2 \sum_{i=1}^{J} \beta_i^2 \left( \frac{\sigma_{0,i}^2}{w_{0,i}} + \frac{\sigma_{1,i}^2}{w_{1,i}} \right)} \quad \text{(Lemma 5)} .$$

$\boldsymbol{w} \in \mathcal{C}_{\text{agnostic}}$ implies $w_{a,i} = \alpha_i u_a$ with $(u_0, \dots, u_K) \in \Sigma_{K+1}$. For this reason,

$$\boldsymbol{w}^\star = \text{argmin}_{\boldsymbol{u}: u_0 + u_1 = 1} \sum_{i=1}^{J} \frac{\beta_i^2}{\alpha_i} \left( \frac{\sigma_{0,i}^2}{u_0} + \frac{\sigma_{1,i}^2}{u_1} \right) .$$

We let $c_a := \sum_{i=1}^{J} \frac{\beta_i^2 \sigma_{a,i}^2}{\alpha_i}$ for $a \in \{0, 1\}$. Plugging $u_1 = 1 - u_0$ in the previous expression and differentiating with respect to $u_0$ brings the condition

$$u_0^2 + 2u_0 \frac{c_0}{c_1 - c_0} - \frac{c_0}{c_1 - c_0} .$$

Solving this polynomial and using that $\boldsymbol{u} \in \Sigma_2$ gives the unique solution

$$u_0^\star = \frac{\sqrt{c_0}}{\sqrt{c_0} + \sqrt{c_1}} .$$

Implying,

$$w_{0,i}^\star = \alpha_i \frac{\sqrt{\sum_{i=1}^{J} \frac{\beta_i^2 \sigma_{0,i}^2}{\alpha_i}}}{\sqrt{\sum_{i=1}^{J} \frac{\beta_i^2 \sigma_{0,i}^2}{\alpha_i}} + \sqrt{\sum_{i=1}^{J} \frac{\beta_i^2 \sigma_{1,i}^2}{\alpha_i}}} \quad \text{and} \quad w_{1,i}^\star = \alpha_i \frac{\sqrt{\sum_{i=1}^{J} \frac{\beta_i^2 \sigma_{1,i}^2}{\alpha_i}}}{\sqrt{\sum_{i=1}^{J} \frac{\beta_i^2 \sigma_{0,i}^2}{\alpha_i}} + \sqrt{\sum_{i=1}^{J} \frac{\beta_i^2 \sigma_{1,i}^2}{\alpha_i}}} .$$

With those values,

$$T^\star_{\text{agnostic}}(\boldsymbol{\mu}) = \frac{2\left(\sqrt{\sum_{i=1}^{J}\frac{\beta_i^2\sigma_{0,i}^2}{\alpha_i}} + \sqrt{\sum_{i=1}^{J}\frac{\beta_i^2\sigma_{1,i}^2}{\alpha_i}}\right)^2}{\Delta_1^2}.$$

**Proportional mode**   Following the same line of proof, gives

$$T^\star_{\text{prop}}(\boldsymbol{\mu})^{-1} = \sup_{\boldsymbol{w}\in\mathcal{C}_{\text{prop}}} \frac{\Delta_1^2}{2\sum_{i=1}^{J}\beta_i^2\left(\frac{\sigma_{0,i}^2}{w_{0,i}} + \frac{\sigma_{1,i}^2}{w_{1,i}}\right)}.$$

The main difference is now on the constraints on the weights. In the proportional mode, following Lemma 3, $\forall i \leq J, \sum_{a=0}^{1} w_{a,i} = \alpha_i$. We consider the Lagrangian function:

$$L(w_0, w_1, q_1, \ldots, q_J) = \sum_{i=1}^{J}\beta_i^2\left(\frac{\sigma_{0,i}^2}{w_{0,i}} + \frac{\sigma_{1,i}^2}{w_{1,i}}\right) + \sum_{i=1}^{J} q_i\left(\sum_{a\in\{0,1\}} w_{a,i} - \alpha_i\right).$$

Differentiating with respect to $w_{0,i}$ and $w_{1,i}$ gives the constraints:

$$\frac{-\beta_i^2\sigma_{0,i}^2}{w_{0,i}^2} + q_i = 0 \quad \text{and} \quad \frac{-\beta_i^2\sigma_{1,i}^2}{w_{1,i}^2} + q_i = 0.$$

From which we can deduce

$$\frac{w_{0,i}}{\sigma_{0,i}} = \frac{w_{1,i}}{\sigma_{1,i}}.$$

From $w_{0,i} + w_{1,i} = \alpha_i$, we deduce,

$$q_i^\star = \frac{\beta_i^2(\sigma_{0,i} + \sigma_{1,i})^2}{\alpha_i^2}.$$

Plugging this value in the first constraint gives

$$w_{0,i}^\star = \alpha_i\frac{\sigma_{0,i}}{\sigma_{0,i} + \sigma_{1,i}} \quad \text{and} \quad w_{1,i}^\star = \alpha_i\frac{\sigma_{1,i}}{\sigma_{0,i} + \sigma_{1,i}}.$$

Using those weights,

$$T^\star_{\text{prop}}(\boldsymbol{\mu}) = \frac{2\sum_{i=1}^{J}\frac{\beta_i^2}{\alpha_i}(\sigma_{0,i} + \sigma_{1,i})^2}{\Delta_1^2}.$$

**Active mode**   Following the proof of Proposition 2, one has

$$T^\star_{\text{active}}(\boldsymbol{\mu})^{-1} = \sup_{\boldsymbol{w}\in\Sigma_{(K+1)J}} \frac{\Delta_1^2}{2\sum_{i=1}^{J}\beta_i^2\left(\frac{\sigma_{0,i}^2}{w_{0,i}} + \frac{\sigma_{1,i}^2}{w_{1,i}}\right)}. \tag{11}$$

Using the constraint $\boldsymbol{w}\in\Sigma_{(K+1)J}$, one gets

$$w_{1,J} = 1 - \sum_{a=\{0,1\}}\sum_{i=1}^{J-1} w_{a,i} - w_{0,J}. \tag{12}$$

We need to minimize the function (where $w_{1,J}$ has been replaced by the expression from Equation 12)

$$f(\boldsymbol{w}) = \sum_{a=\{0,1\}}\sum_{i=1}^{J-1}\beta_i^2\frac{\sigma_{a,j}^2}{w_{a,j}} + \beta_J^2\frac{\sigma_{0,J}^2}{w_{0,J}} + \beta_J^2\frac{\sigma_{1,J}^2}{1 - \sum_{a=0}^{1}\sum_{i=1}^{J-1} w_{a,i} - w_{0,J}}.$$

For $i \leq J - 1$, taking the derivative with respect to $w_{0,i}$ and $w_{1,i}$ gives the following constraints

$$\beta_i^2 \sigma_{0,i}^2 \left( 1 - \sum_{a=0}^{1} \sum_{i=1}^{J-1} w_{a,i} - w_{0,J} \right)^2 = \beta_J^2 \sigma_{1,J}^2 w_{0,i}^2 \ ,$$

$$\beta_j^2 \sigma_{1,i}^2 \left( 1 - \sum_{a=0}^{1} \sum_{i=1}^{J-1} w_{a,i} - w_{0,J} \right)^2 = \beta_J^2 \sigma_{1,J}^2 w_{1,i}^2 \ .$$

From which we deduce

$$\forall i \leq J - 1, \quad \frac{w_{0,i}}{\sigma_{0,i}} = \frac{w_{1,i}}{\sigma_{1,i}} \ . \tag{13}$$

Differentiating with respect to $w_{1,J}$ gives

$$\sigma_{0,J} \left( 1 - \sum_{a=0}^{1} \sum_{i=1}^{J-1} w_{a,i} - w_{0,J} \right) = \sigma_{1,J} w_{0,J} \ .$$

Rearranging and using Equation 13 gives,

$$w_{0,J} = \frac{\sigma_{0,J}}{\sigma_{0,J} + \sigma_{1,J}} - \sum_{i=1}^{J-1} \frac{w_{0,i}}{\sigma_{0,i}} \frac{\sigma_{0,i} + \sigma_{1,i}}{\sigma_{0,J} + \sigma_{1,J}} \sigma_{0,J} \ . \tag{14}$$

Using Equation 13 and Equation 14, we define the function

$$g(w_{0,1}, \ldots, w_{0,J-1}) = \sum_{i=1}^{J-1} \beta_i^2 \frac{\sigma_{0,i}}{w_{0,i}} (\sigma_{0,i} + \sigma_{1,i}) + \frac{\beta_J^2 (\sigma_{0,J} + \sigma_{1,J})^2}{1 - \sum_{i=1}^{J-1} \frac{w_{0,i}(\sigma_{0,i} + \sigma_{1,i})}{\sigma_{0,i}}} \ .$$

Differentiating with respect to $w_{0,i}$ for $i \leq J - 1$ brings

$$\forall i \leq J - 1, \quad \frac{|\beta_i| \sigma_{0,i}}{\sigma_{0,J} + \sigma_{1,J}} \left( 1 - \sum_{i=1}^{J-1} \frac{w_{0,i}}{\sigma_{0,i}} (\sigma_{0,i} + \sigma_{1,i}) \right) = |\beta_J| w_{0,i} \ . \tag{15}$$

Multiplying both sides of this equation by $(\sigma_{0,i} + \sigma_{1,i})/\sigma_{0,i}$ and summing for $i \leq J - 1$,

$$\sum_{i=1}^{J-1} \frac{w_{0,i}}{\sigma_{0,i}} (\sigma_{0,i} + \sigma_{1,i}) = \frac{\sum_{i=1}^{J-1} |\beta_i|(\sigma_{0,i} + \sigma_{1,i})}{\sum_{i=1}^{J} |\beta_i|(\sigma_{0,i} + \sigma_{1,i})} \ .$$

Plugging this value back in Equation 15 one has:

$$\forall i \leq J - 1, \quad w_{0,i} = \frac{|\beta_i| \sigma_{0,i}}{\sum_{i=1}^{J} |\beta_i|(\sigma_{0,i} + \sigma_{1,i})} \ .$$

From Equation 13, we deduce,

$$\forall i \leq J - 1, \quad w_{1,i} = \frac{|\beta_i| \sigma_{1,i}}{\sum_{i=1}^{J} |\beta_i|(\sigma_{0,i} + \sigma_{1,i})} \ .$$

We obtain the value of $w_{0,J}$ using Equation 14 and that of $w_{1,J}$ using Equation 12. Plugging those weights in the expression given by Equation 11 yields the characteristic time. $\qquad \square$

# D  General form of the optimal allocation in the Gaussian case

**Proposition 3** (Efficient computation in the Gaussian case)**.** *With Gaussian distributions with a known variance $\sigma^2$, letting $(u_0^\star, \ldots, u_K^\star) = \mathrm{argmax}_{u \in \Sigma_{K+1}} \min_{b \neq 0} \frac{\Delta_b^2}{2 \left( \frac{1}{u_0} + \frac{1}{u_b} \right)}$, the optimal weights for the active mode satisfy*

$$\forall a \in \{0, \ldots, K\}, \ \forall i \leq J, \ w_{a,i}^\star = u_a^\star \frac{|\beta_i|}{\sum_{i=1}^{J} |\beta_i|} \ .$$

*If, in addition $\boldsymbol{\alpha} = \boldsymbol{\beta}$, the above also holds for the agnostic and the proportional modes.*

*Proof.* From Lemma 5, when the distribution are Gaussian with a known variance $\sigma^2$ one has

$$T_{\text{active}}^\star(\boldsymbol{\mu})^{-1} = \sup_{\boldsymbol{w}\in\Sigma_{(K+1)J}} \min_{b\neq 0} \frac{\Delta_1^2}{2\sum_{i=1}^J \beta_i^2 \left(\frac{\sigma^2}{w_{0,i}} + \frac{\sigma^2}{w_{b,i}}\right)} \ .$$

Using the same continuity argument than in [10], we know that the supremum of $\boldsymbol{w}$ is attained and is indeed a maximum. Let

$$\Lambda_b(v,w) := \frac{\Delta_b^2}{2\sum_{i=1}^J \beta_i^2 \left(\frac{\sigma^2}{v_i} + \frac{\sigma^2}{w_i}\right)} \ .$$

Then

$$\max_{\boldsymbol{w}\in\Sigma_{(K+1)J}} \min_{b\neq 0} \Lambda_b(w_0,w_b) = \max_{\substack{u\in\Sigma_{K+1} \\ \forall a, \sum_{i=1}^J w_{a,i}=u_a}} \min_{b\neq 0} \Lambda_b(w_0,w_b)$$

$$= \max_{u\in\Sigma_{K+1}} \max_{\substack{w\in\Sigma_{(K+1)J} \\ \forall a, \sum_i w_{a,i}=u_a}} \min_{b\neq 0} \Lambda_b(w_0,w_b)$$

$$\leq \max_{u\in\Sigma_{K+1}} \min_{b\neq 0} \max_{\substack{w\in\Sigma_{(K+1)J} \\ \forall a, \sum_i w_{a,i}=u_a}} \Lambda_b(w_0,w_b) \quad \text{(Max-min inequality)} \ .$$

Let $b \neq 0$,

$$\max_{\substack{w\in\Sigma_{(K+1)J} \\ \forall a, \sum_i w_{a,i}=u_a}} \Lambda_b(w_0,w_b) = \max_{\substack{w\in\Sigma_{(K+1)J} \\ \forall a, \sum_i w_{a,i}=u_a}} \frac{\Delta_b^2}{2\sum_{i=1}^J \beta_i^2 \left(\frac{\sigma^2}{w_{0,i}} + \frac{\sigma^2}{w_{b,i}}\right)} \ .$$

Equivalently, we are interested in

$$\min_{\substack{w\in\Sigma_{(K+1)J} \\ \forall a, \sum_i w_{a,i}=u_a}} \sum_{i=1}^J \beta_i^2 \left(\frac{\sigma^2}{w_{0,i}} + \frac{\sigma^2}{w_{b,i}}\right) \ .$$

We introduce the associated Lagrangian function

$$f(w,q) = \sum_{i=1}^J \beta_i^2 \left(\frac{1}{w_{0,i}} + \frac{1}{w_{b,i}}\right) + \sum_{a=0}^K q_a \left(\sum_{i=1}^J w_{a,i} - u_a\right) \ .$$

Taking the derivative with respect to $w_{0,i}$ and $w_{b,i}$ for the different values of $i$ yields

$$w_{0,i} = \frac{|\beta_i|}{\sqrt{q_0}} \quad \text{and} \quad w_{b,i} = \frac{|\beta_i|}{\sqrt{q_b}} \ .$$

Summing over $i$ implies that

$$\sqrt{q_0} = \frac{\sum_{i=1}^J |\beta_i|}{u_0} \quad \text{and} \quad \sqrt{q_b} = \frac{\sum_{i=1}^J |\beta_i|}{u_b} \ ,$$

and plugging the above in the expression of the weights yields

$$w_{0,i} = \frac{|\beta_i|}{\sum_{i=1}^J |\beta_i|} u_0 \quad \text{and} \quad w_{b,i} = \frac{|\beta_i|}{\sum_{i=1}^J |\beta_i|} u_b \ .$$

In particular,

$$\max_{\substack{w\in\Sigma_{(K+1)J} \\ \forall a, \sum_i w_{a,i}=u_a}} \frac{\Delta_b^2}{2\sum_{i=1}^J \beta_i^2 \left(\frac{\sigma^2}{w_{0,i}} + \frac{\sigma^2}{w_{b,i}}\right)} = \frac{\Delta_b^2}{2\sigma^2 \left(\sum_{i=1}^J |\beta_i|\right)^2 \left(\frac{1}{u_0} + \frac{1}{u_b}\right)} \ , \tag{16}$$

yielding

$$\max_{\boldsymbol{w}\in\Sigma_{(K+1)J}} \min_{b\neq 0} \frac{\Delta_b^2}{2\sum_{i=1}^J \beta_i^2 \left(\frac{\sigma^2}{w_{0,i}} + \frac{\sigma^2}{w_{b,i}}\right)} \leq \frac{1}{2\sigma^2 \left(\sum_{i=1}^J |\beta_i|\right)^2} \max_{u\in\Sigma_{(K+1)J}} \min_{b\neq 0} \frac{\Delta_b^2}{\left(\frac{1}{u_0} + \frac{1}{u_b}\right)} \ .$$

On the other hand, letting $w_{a,i} = \frac{|\beta_i|}{\sum_{i=1}^{J} |\beta_i|} u_a$ with $\sum_a u_a = 1$ we have

$$\frac{1}{2\sigma^2 \left(\sum_{i=1}^{J} |\beta_i|\right)^2} \max_{u \in \Sigma_{(K+1)J}} \min_{b \neq 0} \frac{\Delta_b^2}{\left(\frac{1}{u_0} + \frac{1}{u_b}\right)} \leq \max_{w \in \Sigma_{(K+1)J}} \min_{b \neq 0} \frac{\Delta_b^2}{2 \sum_{i=1}^{J} \beta_i^2 \left(\frac{\sigma^2}{w_{0,i}} + \frac{\sigma^2}{w_{b,i}}\right)} \, ,$$

showing that the two optimization programs are equivalent and that when denoting

$$(u_0^\star, \ldots, u_K^\star) = \operatorname{argmax}_{u \in \Sigma_{(K+1)J}} \min_{b \neq 0} \frac{\Delta_b^2}{\left(\frac{1}{u_0} + \frac{1}{u_b}\right)} \, ,$$

one has

$$\forall a \in \{0, \ldots, K\}, \ \forall i \leq J, \ w_{a,i}^\star = u_a^\star \frac{|\beta_i|}{\sum_{i=1}^{J} |\beta_i|} \, .$$

This corresponds to the optimal allocation strategy in the *active* mode. Recalling that when $\alpha = \beta$, the optimal weights for the *active* mode satisfy both $\mathcal{C}_{\text{prop}}$ and $\mathcal{C}_{\text{agnostic}}$ completes the proof. $\qquad\square$

# E  Asymptotic Optimality: Proof of Theorem 2

In this section we show that T-a-S with C-tracking and a certain threshold $\beta(t, \delta)$ is safely calibrated and asymptotically optimal. This is an important sanity check to validate our approach theoretically. Note that for the experimental validation we have explored a practically appealing variant of this algorithm: we employ an iterative scheme to approximate $w^*(\hat{\mu}(t))$, use D-tracking, and stylise the threshold.

Safe calibration follows from the definition of the recommendation rule (we report the answer $\mathcal{S}_\beta(\hat{\mu}(t))$ at the empirical estimate $\hat{\mu}(t)$ of the bandit instance), together with the computation of the risk assessment $\hat{\delta}_t$. It does not depend on the sampling rule. Our confidence level $\hat{\delta}_t$ is obtained by inverting the threshold $\beta(t, \delta)$ at the GLR statistic (7). Safe calibration then follows from an anytime-valid GLR deviation inequality with boundary $\beta(t, \delta)$. We refer to [13, Proposition 23] for a boundary that is, in case of the ABC-S problem, of order $\ln \frac{1}{\delta} + K + 2J \cdot O(\ln \ln \frac{t}{\delta})$.

It remains to argue that the T-a-S sampling rule converges to the oracle weights. The original T-a-S proof for the BAI problem is due to [10, Theorem 14]. An upgrade to any single-answer problem, including our ABC-S, is due to [6]. For active mode, their theorem applies directly, while for agnostic mode it applies with the pair $(I_t, X_t)$ regarded as the observation. We get:

**Theorem 3** ([6, Theorems 7 and 10]). *For all ABC-S instances $\mu \in \mathcal{L}$ in active mode and agnostic mode, Track-and-Stop with C-tracking and stopping threshold $\beta(t, \delta) = \ln(t^2/\delta) + O(1)$ is $\delta$-correct with asymptotically optimal sample complexity.*

In *proportional* mode, we have the additional constraint that the learner chooses its arm in response to seeing (but not controlling) the subpopulation $I_t$. Still, the tracking convergence result [6, Lemma 6] goes through, upon observing that the empirical distribution of $I_t$ converges to $\alpha$ by the law of the large numbers, and hence our conditional tracking (see "sampling rule" in Section 3) adds the right conditional to the right marginal. All in all, the computed joint weights converge to the joint $w_{\text{prop}}^*(\mu)$, and tracking makes the sampling proportions also converge there.

We conclude with a remark on our use of D-tracking. Recall that D-tracking is the idea of advancing $N_a(t)$ towards $t$ times the most current oracle weights, i.e. $tw_a^*(\hat{\mu}(t))$, while C-tracking makes $N_a(t)$ advance towards the sum of encountered oracle weights, i.e. $\sum_{s=1}^{t} w_a^*(\hat{\mu}(s))$. As argued in [7, Appendix E], D-tracking can fail to make $N_a(t)/t$ converge to $w_a^*(\mu)$. However, this requires that the maximiser of the lower bound problem is not unique at $\mu$ (as we are maximising a concave function, the set of maximisers is always convex). Here we argue that such a situation does not occur for the ABC-S problem. To see why, we argue that the lower bound objective, as a function of $w$, is strictly concave. It suffices to show this for the *active* mode problem, as the problems for the other modes are further constrained maximisation problems of the same objective.

**Lemma 6.** *Fix a bandit instance $\mu \in \mathcal{L}$. Let $\lambda \mapsto d(\mu_{a,j}, \lambda)$ be a strongly convex function for each arm $a$ and subpopulation $j$. Then for the ABC-S problem with $\beta$ such that $\beta_j \neq 0$ for all $j$, the oracle weights $w^*(\mu)$ are unique.*

*Proof.* Let $\boldsymbol{w}^*(\boldsymbol{\mu})$ be any oracle weights at $\boldsymbol{\mu}$. We will show the lower bound objective (4) is strictly concave as a function of $\boldsymbol{w}$ around $\boldsymbol{w}^*(\boldsymbol{\mu})$, so that $\boldsymbol{w}^*(\boldsymbol{\mu})$ was in fact unique. For each $k > 0$, let $\boldsymbol{\lambda}^k$ be the minimiser in $\mathrm{Alt}^k(\boldsymbol{\mu})$ of the weighted divergence in (4).

We perform a second-order Taylor expansion of the inner objective around $\boldsymbol{\lambda}^k$, which is a good approximation near $\boldsymbol{\lambda}^k$ (which is, after all, what matters when reasoning about $\boldsymbol{w}$ near $\boldsymbol{w}^*(\boldsymbol{\mu})$). To this end, let us abbreviate the divergences, and their first and second derivatives in their second argument by $d_{aj}^k := d(\mu_{a,j}, \lambda_{a,j}^k)$, $g_{aj}^k := d'(\mu_{a,j}, \lambda_{a,j}^k)$ and $h_{aj}^k := d''(\mu_{a,j}, \lambda_{a,j}^k)$, which all depend on $\boldsymbol{\lambda}^k$. A second-order Taylor expansion of the inner objective of (4) around $\boldsymbol{\lambda}^k$ yields

$$
\inf_{\boldsymbol{\lambda} \in \mathrm{Alt}_k} \sum_{a,j} w_{a,j} d(\mu_{a,j}, \lambda_{a,j}) \approx \sum_{a \in \{0,k\}, j} w_{a,j} \left( d_{aj}^k - \frac{(g_{aj}^k)^2}{2 h_{aj}^k} \right) + \frac{\left( \sum_j \beta_j \left( \frac{g_{0j}^k}{h_{0j}^k} - \frac{g_{kj}^k}{h_{kj}^k} \right) \right)^2}{2 \sum_{a \in \{0,k\}, j} \frac{\beta_j^2}{w_{a,j} h_{aj}^k}}
$$

where the optimiser is given by

$$
\lambda_{a,j} = \lambda_{a,j}^k - \frac{g_{aj}^k}{h_{aj}^k} + \frac{\beta_j (\delta_{a=0} - \delta_{a=k})}{w_{a,j} h_{aj}^k} \frac{\sum_j \beta_j \left( \frac{g_{0j}^k}{h_{0j}^k} - \frac{g_{kj}^k}{h_{kj}^k} \right)}{\sum_{a \in \{0,k\}, j} \frac{\beta_j^2}{w_{a,j} h_{aj}^k}} .
$$

Due to the last term, each of these is a strictly concave function of $w_{a,j}$ for $a \in \{0, k\}$ and all $j \leq J$ (here we use $\beta_j \neq 0$ and strong convexity $h_{aj}^k > 0$).

Now we still need to consider the $\max_{\boldsymbol{w} \in \Sigma_{(K+1) \times J}} \min_{k>0}$ problem. Let's convexify this for the inside finite min, and min-max swap to get a problem of the form $\min_{\boldsymbol{q} \in \Sigma_K} \max_{\boldsymbol{w} \in \Sigma_{(K+1) \times J}}$. Fixing the minimax outer strategy for $\boldsymbol{q}$, we find that $\boldsymbol{w}$ is the maximiser of the strictly concave function

$$
\boldsymbol{w} \mapsto \sum_{k>0} q_k \left( \sum_{a \in \{0,k\}, j} w_{a,j} \left( d_{aj}^k - \frac{(g_{aj}^k)^2}{2 h_{aj}^k} \right) + \frac{\left( \sum_j \beta_j \left( \frac{g_{0j}^k}{h_{0j}^k} - \frac{g_{kj}^k}{h_{kj}^k} \right) \right)^2}{2 \sum_{a \in \{0,k\}, j} \frac{\beta_j^2}{w_{a,j} h_{aj}^k}} \right)
$$

To complete the argument, we argue that $q_k > 0$ for all $k > 0$, or, equivalently, that at $\boldsymbol{w}^*$ the $\min_{k>0}$ are all equalised. For if not, we can move mass from $w_{k,j}$ for the higher $k > 0$ to $w_{k',j}$ for the lower $k'$ and increase the objective value. This then proves that $\boldsymbol{w}^*(\boldsymbol{\mu})$ is unique, as the objective function is bounded above by a strictly concave function itself maximised at $\boldsymbol{w} = \boldsymbol{w}^*(\boldsymbol{\mu})$. $\qquad\square$

## F  Algorithm Details

In this section we go into more details on the algorithm for each mode. Let us start with some notation. Let $\beta(\delta, t)$ be a threshold function. We denote the inverse of $\beta(t, \delta)$ in its second argument by

$$
\beta^{-1}(t, \Lambda) = \min \{ \delta \in (0, 1) | \Lambda \geq \beta(t, \delta) \} .
$$

We extend the definition of the GLR statistic to sample frequencies $\boldsymbol{w}$ and bandit $\boldsymbol{\mu}$ by

$$
\Lambda(\boldsymbol{w}, \boldsymbol{\mu}) := \min_{b \neq 0} \inf_{\boldsymbol{\lambda} \in \mathcal{L} : \lambda_0 = \lambda_b} \sum_{a \in \{0,b\}} \sum_{i=1}^{J} w_{a,i} d(\mu_{a,i}, \lambda_{a,i}) ,
$$

so that the original definition (7) is $\Lambda(t) = \Lambda(\boldsymbol{N}(t)/t, \hat{\boldsymbol{\mu}}(t))$. For any $\boldsymbol{\mu}$, we denote by $\nabla_{\boldsymbol{w}} \Lambda(\boldsymbol{w}, \boldsymbol{\mu})$ any sub-gradient of $\boldsymbol{w} \mapsto \Lambda(\boldsymbol{w}, \boldsymbol{\mu})$. We can obtain one such a sub-gradient by letting $(b, \boldsymbol{\lambda})$ be any minimiser of $\Lambda(\boldsymbol{w}, \boldsymbol{\mu})$, and constructing the vector with entry $(a, i)$ given by

$$
(a, i) \mapsto \begin{cases} d(\mu_{a,i}, \lambda_{a,i}) & \text{if } a \in \{0, b\} \\ 0 & \text{otherwise} \end{cases} .
$$

Our algorithms will make use of an online learning method (called $\mathcal{A}$ below) for linear losses defined on the simplex. This online learning task is known as the Hedge or Experts setting in the literature.

We will make use of AdaHedge [5], as it adapts automatically to the range of the losses and does not require tuning. Our methods for the active, proportional and agnostic modes are displayed as Algorithms 1, 2 and 3. Each algorithm consists of a Forced Exploration part, which serves to ensure that the empirical estimate of the bandit model converges, i.e. $\hat{\boldsymbol{\mu}}(t) \to \boldsymbol{\mu}$. By forcing exploration sublinearly often, the main term in the sample complexity is unaffected asymptotically. Each algorithm further makes use of online learning to compute $\boldsymbol{w}^*(\boldsymbol{\mu})$. In the notation of this section, we have

$$\boldsymbol{w}^*(\boldsymbol{\mu}) \;=\; \operatorname*{argmax}_{\boldsymbol{w} \in \mathcal{C}} \Lambda(\boldsymbol{w}, \boldsymbol{\mu}) \;.$$

Our approach to learning $\boldsymbol{w}^*(\boldsymbol{\mu})$ is to perform gradient steps on the plug-in loss function $\boldsymbol{w} \mapsto -\Lambda(\boldsymbol{w}, \hat{\boldsymbol{\mu}}(t))$. It is in the convex domain $\mathcal{C} \subseteq \Sigma_{(K+1) \times J}$ that we see the main difference between the three modes. Recall from Section 2.3 that in the active mode $\boldsymbol{w}$ is not constrained further, in the proportional mode the subpopulation marginal of $\boldsymbol{w}$ must equal $\boldsymbol{\alpha}$, i.e. $\langle \mathbf{1}, \boldsymbol{w} \rangle = \boldsymbol{\alpha}$, and in the agnostic mode $\boldsymbol{w}$ must be the independent product $\boldsymbol{w} = \boldsymbol{v}\boldsymbol{\alpha}$ of some arm marginal $\boldsymbol{v} \in \Sigma_{K+1}$ and the subpopulation frequencies $\boldsymbol{\alpha}$. We hence need to design online learners for each of the three $\mathcal{C}$. In the active case, we have one learner $\mathcal{A}$ that learns the full joint $\boldsymbol{w}^*(a, j)$ directly, in the proportional case we use one learner $\mathcal{A}_j$ for each subpopulation $j \in [J]$ to learn the conditional distribution $\boldsymbol{w}^*(a|j)$, and in the agnostic case we again use one learner to learn the common marginal $\boldsymbol{w}^*(a)$. This difference is reflected in the loss function used in each mode, and hence in the gradient that is fed to each learner. In the active case we use the full $(K+1) \times J$ gradients

$$\boldsymbol{\ell}_t^{\text{active}} \;:=\; -\nabla_{\boldsymbol{w}} \Lambda(\boldsymbol{w}_t, \hat{\boldsymbol{\mu}}(t)) \;.$$

In the proportional case we have $\boldsymbol{w}(a, i) = \boldsymbol{w}(a|i)\alpha_i$, and by the chain rule we hence have gradients

$$\boldsymbol{\ell}_t^{i, \text{ proportional}} \;:=\; -\nabla_{\boldsymbol{w}(a|i)} \Lambda(\boldsymbol{w}_t, \hat{\boldsymbol{\mu}}(t)) \;=\; -\alpha_i \nabla_{\boldsymbol{w}} \Lambda(\boldsymbol{w}_t, \hat{\boldsymbol{\mu}}(t)) \boldsymbol{e}_i \;.$$

Finally, in the agnostic case we have $\boldsymbol{w}(a, i) = w(a)\alpha_i$, and again by the chain rule we have

$$\boldsymbol{\ell}_t^{\text{agnostic}} \;:=\; -\nabla_{\boldsymbol{w}(a)} \Lambda(\boldsymbol{w}_t, \hat{\boldsymbol{\mu}}(t)) \;=\; -\nabla_{\boldsymbol{w}} \Lambda(\boldsymbol{w}_t, \hat{\boldsymbol{\mu}}(t)) \boldsymbol{\alpha} \;.$$

**Run Time** In each of the three modes, our algorithms evaluate $\Lambda$ for the confidence in the recommendation, compute one sub-gradient of $\Lambda$ for the loss function, and spend $O(K \times J)$ time bookkeeping. Evaluation and sub-gradient computation for $\Lambda$ boil down to solving a convex minimisation problem with an equality constraint. We use Newton's method with backtracking line search to find the minimiser given $b$. Each Newton iteration takes $O(J^2)$ time (recall that only 2 arms are involved), and we never needed more than 40. Doing this $K$ times for the explicit minimum over $b$ yields a total per iteration run time of $O(KJ^2)$.

---

**Algorithm 1** Algorithm for Active Mode.

---

**Require:** Online learner $\mathcal{A}$ for $(K+1) \times J$ experts.
    **for** $t = 1, 2, \ldots$ **do**
        **if** any pair $(a, i)$ has $N_{a,i}(t-1) \le \sqrt{t}$ **then**
            Pick $A_t, I_t$ any such pair                                       ▷ Forced Exploration
            Obtain sample $X_t$ from $\nu_{A_t, I_t}$.
        **else**
            Get $\boldsymbol{w}_t$ from online learner $\mathcal{A}$
            Pick $(A_t, I_t) \in \operatorname{argmin}_{a,i} N_{a,i}(t-1) - t w_t(a, i)$         ▷ Direct Tracking
            Obtain sample $X_t$ from $\nu_{A_t, I_t}$.
            Send loss vector $\boldsymbol{\ell}_t = -\nabla_{\boldsymbol{w}} \Lambda(\boldsymbol{w}_t, \hat{\boldsymbol{\mu}}(t))$ to $\mathcal{A}$
        **end if**
        Recommend $\hat{\mathcal{S}}_t = \mathcal{S}_{\boldsymbol{\beta}}(\hat{\boldsymbol{\mu}}(t))$ at confidence $\delta_t = \beta^{-1}(t, \Lambda(\boldsymbol{N}(t)/t, \hat{\boldsymbol{\mu}}(t)))$.
    **end for**

---

 

---

**Algorithm 2** Algorithm for Proportional Mode.

---

**Require:** $J$ online learners $\mathcal{A}^{(1)}, \ldots, \mathcal{A}^{(J)}$ for $(K+1)$ experts each.
    **for** $t = 1, 2, \ldots$ **do**
        See $I_t \sim \boldsymbol{\alpha}$.
        **if** any arm $a$ has $N_{a,I_t}(t-1) \le \sqrt{\sum_a N_{a,I_t}(t-1)}$ **then**
            Pick $A_t$ any such arm                                   ▷ Forced Exploration
            Obtain sample $X_t$ from $\nu_{A_t, I_t}$.
        **else**
            Get $\boldsymbol{w}_t^{(j)}$ from each online learner $\mathcal{A}^{(j)}$
            Pick $A_t \in \operatorname{argmin}_a N_{a,I_t}(t-1) - t w_t^{(I_t)}(a)$         ▷ Direct Tracking
            Obtain sample $X_t$ from $\nu_{A_t, I_t}$.
            For $j \in [J]$, send loss vector $\boldsymbol{\ell}_t^{(j)} = -\alpha_j \nabla_{\boldsymbol{w}} \Lambda \left( [\alpha_1 \boldsymbol{w}_t^{(1)} \cdots \alpha_J \boldsymbol{w}_t^{(J)}], \hat{\boldsymbol{\mu}}(t) \right) \boldsymbol{e}_j$ to $\mathcal{A}^{(j)}$
        **end if**
        Recommend $\hat{\mathcal{S}}_t = \mathcal{S}_{\boldsymbol{\beta}}(\hat{\boldsymbol{\mu}}(t))$ at confidence $\delta_t = \beta^{-1}(t, \Lambda(\boldsymbol{N}(t)/t, \hat{\boldsymbol{\mu}}(t)))$.
    **end for**

---

 

---

**Algorithm 3** Algorithm for Agnostic Mode.

---

**Require:** Online learner $\mathcal{A}$ for $(K+1)$ experts.
    **for** $t = 1, 2, \ldots$ **do**
        **if** any arm $a$ has $\sum_{j=1}^{J} N_{a,j}(t-1) \le \sqrt{t}$ **then**
            Pick $A_t$ any such arm                                   ▷ Forced Exploration
            See $I_t \sim \boldsymbol{\alpha}$.
            Obtain sample $X_t$ from $\nu_{A_t, I_t}$.
        **else**
            Get $\boldsymbol{w}_t$ from online learner $\mathcal{A}$
            Pick $A_t \in \operatorname{argmin}_a \sum_{j=1}^{J} N_{a,j}(t-1) - t w_t(a)$         ▷ Direct Tracking
            Obtain sample $X_t$ from $\nu_{A_t, I_t}$.
            See $I_t \sim \boldsymbol{\alpha}$.
            Send loss vector $\boldsymbol{\ell}_t = -\nabla_{\boldsymbol{w}} \Lambda \left( \boldsymbol{w}_t \boldsymbol{\alpha}^\top, \hat{\boldsymbol{\mu}}(t) \right) \boldsymbol{\alpha}$ to $\mathcal{A}$.
        **end if**
        Recommend $\hat{\mathcal{S}}_t = \mathcal{S}_{\boldsymbol{\beta}}(\hat{\boldsymbol{\mu}}(t))$ at confidence $\delta_t = \beta^{-1}(t, \Lambda(\boldsymbol{N}(t)/t, \hat{\boldsymbol{\mu}}(t)))$.
    **end for**

---

# G Details of A/B/n experiment

|   | 1 | 2 | 3 | 4 |
|---|------|------|------|------|
| 0 | 0.0296 | 0.0372 | 0.0588 | 0.0620 |
| 1 | 0.0300 | 0.0373 | 0.0596 | 0.0626 |
| 2 | 0.0295 | 0.0373 | 0.0591 | 0.0630 |

(a) Estimated click probabilities for the different options and seasons

|   | 1 |
|---|------|
| 0 | 0.1958 |
| 1 | 0.2950 |
| 2 | 0.2813 |
| 3 | 0.2279 |

(b) frequency and importance vector ($\alpha = \beta$)

|   | 1 | 2 | 3 | 4 |
|---|------|------|------|------|
| 0 | 0.0719 | 0.1222 | 0.1311 | 0.1238 |
| 1 | 0.0215 | 0.0475 | 0.0340 | 0.0352 |
| 2 | 0.0614 | 0.1060 | 0.1377 | 0.1078 |

(c) $w^*$ for *active*, $T^* = 3.98 \cdot 10^6$

|   | 1 | 2 | 3 | 4 |
|---|------|------|------|------|
| 0 | 0.0740 | 0.1246 | 0.1476 | 0.1226 |
| 1 | 0.0108 | 0.0179 | 0.0214 | 0.0178 |
| 2 | 0.0727 | 0.1228 | 0.1460 | 0.1218 |

(d) Sampling proportions *active*

|   | 1 | 2 | 3 | 4 |
|---|------|------|------|------|
| 0 | 0.0814 | 0.1269 | 0.1289 | 0.0889 |
| 1 | 0.0230 | 0.0355 | 0.0457 | 0.0277 |
| 2 | 0.0914 | 0.1326 | 0.1067 | 0.1113 |

(e) $w^*$ for *proportional*, $T^* = 4.06 \cdot 10^6$

|   | 1 | 2 | 3 | 4 |
|---|------|------|------|------|
| 0 | 0.0912 | 0.1374 | 0.1307 | 0.1056 |
| 1 | 0.0148 | 0.0223 | 0.0214 | 0.0173 |
| 2 | 0.0898 | 0.1353 | 0.1293 | 0.1048 |

(f) Sampling proportions *proportional*

|   | 1 |
|---|------|
| 0 | 0.44482 |
| 1 | 0.11111 |
| 2 | 0.44406 |

(g) $w^*$ for *agnostic*, $T^* = 4.61 \cdot 10^6$

|   | 1 |
|---|------|
| 0 | 0.4648 |
| 1 | 0.0766 |
| 2 | 0.4587 |

(h) Sampling proportions *agnostic*

|   | 1 |
|---|------|
| 0 | 0.44480 |
| 1 | 0.11111 |
| 2 | 0.44409 |

(i) $w^*$ for *oblivious*, $T^* = 4.63 \cdot 10^6$

|   | 1 |
|---|------|
| 0 | 0.4647 |
| 1 | 0.0764 |
| 2 | 0.4589 |

(j) Sampling proportions *oblivious*

|   | 1 |
|---|------|
| 0 | 0.500 |
| 1 | 0.076 |
| 2 | 0.424 |

(k) Sampling proportions BC-ABC

Figure 4: Summary of oracle weights and sampling proportions for A/B/n experiment