# OpenReview forum: "A/B/n Testing with Control in the Presence of Subpopulations"
_NeurIPS.cc/2021/Conference — NeurIPS 2021 Poster_

### Official Review · Reviewer_4N5J · 2021-07-04

**Rating:** 5
**Confidence:** 4

**Summary:**

The paper studies a variation of the multi-armed bandit problem, in which one seeks to identify all arms that are better than the control arm, as opposed to the standard “best arm identification” problem. The authors study the sampling complexity and investigate the impact of different modes of interactions in sampling from different arms and subpopulations.

**Ethical Concerns:**

No ethical concerns.

**Main Review:**

- The paper is well-written and the problem is interesting.

- The paper is mathematically rigorous.

- A major concern is that, apart from the modes of interaction considered in this work (which is a good addition), the main result of the paper in Theorem 1 is well-known. The problem is an instance of a classic problem in statistics on optimal design of experiments, initially studied by Chernoff in his seminal work on "sequential design of experiments". Chernoff analyzed the expected stopping time and obtained a very similar expression, which can be viewed as the valuation of a zero-sum game in which one player (the controller) chooses a randomized strategy over the actions (here, the coefficients w in the supremum in (3) for the arms and subpopulations) to maximize the weighted KL distance, while the other player (nature) chooses the worst-case alternative hypothesis (here, the mean vector \lambda) to minimize the weighted KL distance. I believe the main novelty lies in studying different modes of interactions as special cases of this problem.

- From an asymptotic analysis standpoint (the focus of this work), I do not see what the major differences between the BAI problem and the problem at hand are. Consider the scenario in which one starts off identifying the best arm, after which we stop sampling from this arm. The process is repeated to identify the best arm from the remaining ones. After at most K times, we can identify all arms that are better than the control. This is just a constant K factor, which will not change the order complexity.

- The authors do not cite important related work, including the work of Chernoff mentioned earlier on sequential design of experiments, and the work of Lai and Robbins on index policies for multi-armed bandits. There is a large body of closely related work on the topic that the authors ignore (or are probably unaware of).


**Time Spent Reviewing:**

4 hours

---

> ### Author Response · Authors · 2021-08-10
> **Response to Reviewer 4N5J**
>
> We think that there is a misunderstanding of the setting. There are several major differences between the problem considered in this work and standard best arm identification (BAI). First, depending on the interaction mode, the learner has different constraints for the subpopulation choices. Second, instead of identifying the best arm, the objective is to identify all the arms that are better than a control arm (whose performance is also unknown). While some results are based on existing works, obtaining guarantees when adding the subpopulations constraints is far from obvious.
>
> Reviewer 4 suggests to use a BAI algorithm to identify the best arm, then the second-best and so on until the best remaining arm is the control. This simple reduction to BAI is unfortunately extremely inefficient - possibly much more than a factor K, which would already be prohibitive when considering that we target the best sample complexity up to a factor arbitrary close to 1 for delta small enough. In fact, in the case where the control has a lower mean than 2 other arms with equal expectations, the strategy suggested by the reviewer would never stop. Even a modified strategy coupling the BAI sampling rule with a better stopping rule would dramatically undersample the control arm with respect to what an efficient algorithm needs to do (except in the very special case of A/B testing where K=2).
>
> We definitely agree on the importance of the visionary works of Chernoff in the 1950s on sequential design of experiments and sequential test. We can add references to these works if needed, however these are already cited by the papers that we cite, for instance [1] who choose to name the stopping rule that they proposed "Chernoff's stopping rule" (the stopping rule we use in our own work might be seen as an extension of this proposal). We would also be happy to cite the work of Lai and Robbins (one of the most important among the thousands of papers on bandit models) but index policies and regret minimization are topics really distinct from ours, and we do not understand how this citation could help the reader in any way. We prefer to cite [5], a good entry point to the modern study of bandit models which contains all general references, together with articles directly related to our work.
>
> In light of the above comments, we think that the sentence "There is a large body of closely related work on the topic that the authors ignore (or are probably unaware of)" is somewhat contemptuous and factually incorrect. But most importantly the claim that "the main result of the paper in Theorem 1 is well-known" is also incorrect: we are completely unaware of a work by Chernoff or others that would solve the problem treated in our paper (find all arms that are better than a control in the presence of subpopulations) even in a reminiscent way (and we are not talking here of course of basic BAI): please provide a reference and we will be happy to discuss it.
>
> ## References
>
> [1] Garivier et al., "Optimal Best Arm Identification with Fixed Confidence", COLT 2016
>
> [5] Lattimore et al, "Bandits Algorithms", Cambridge Press.

---

### Official Review · Reviewer_x5R6 · 2021-07-15

**Rating:** 7
**Confidence:** 3

**Summary:**

This paper considers the problem of A/B/n testing with subpopulations. At each time step, a sample appears from one of $J$ subpopulations and the player chooses an arm to play. Depending on the variant of the game, the sample may be chosen by the player, drawn by nature and observed before (or after) an arm is chosen, or not observed at all. The goal is to identify the arms whose true means, averaged across the subpopulations according to some known weight vector beta, exceed that of the *control* arm. This places the work in a similar vein as work that identifies all arms above a threshold, but the threshold is defined as the mean of the control arm, which must be estimated.

The paper proposes a subpopulation-aware version of the Track-and-Stop algorithm that allocates samples according to an optimization problem present in the lower bound. Since that optimization problem is somewhat opaque, the paper also derives and discusses the closed-form update rule in the Gaussian setting. The paper concludes with simulated and real-data experiments on Bernoulli observations.

**Limitations And Societal Impact:**

I was surprised that, in the paper checklist, the authors responded N/A to the question of societal impacts. In a work on optimizing across subpopulations, where the paper is motivated with examples from e-commerce (line 21) and clinical trials (line 77), the intersection with questions of societal impact is quite large. For example, the goal of this work is to find arms with large beta-weighted rewards, where a common choice is beta=alpha, the population proportions. Are there settings in which we might choose beta differently from alpha, to satisfy certain fairness constraints? Settings in which we might not want to beta-weight our rewards at all, but rather, say, maximize the minimum reward across populations? Populations with more variance are sampled more than their proportion alpha - are there situations in which this could be good or bad for that population?

It's okay if the authors don't want to discuss these questions, but it's surprising that they indicated societal impact was not applicable to their work.

**Main Review:**

Positives:
- The paper demonstrates a firm grasp of the literature upon which it builds, and the algorithmic and proof techniques therein.
- The paper provides a thorough treatment of different observation settings the player might encounter (active, proportional, agnostic, and oblivious). In addition, the separation of \alpha from \beta provide another layer of detail that makes the setting more broadly applicable.
- Including the optimal weight vector and resulting stopping time for the Gaussian case really helps the reader understand the improvements possible in different settings.

Areas for improvement:
- The algorithm description did not contain enough detail to reproduce the results in the paper, which is even more problematic because code was not submitted. In particular, line 270 (the generalized likelihood ratio test statistic) is itself an optimization problem, and it is not immediately clear how to generate the threshold. If this is trivial, perhaps include it in the appendix with a reference in the main body. Additionally, I suspect that the phrase "(a suitable approximation of)" in lines 259-260 refers to the iterative saddle point solver mentioned in lines 251-252, but it would be nice to see this made explicit.
- After reading the paper, it was unclear whether the method can be practically run for other members of the exponential family. The experiments were run in the Bernoulli setting, but since there wasn't a lot of detail on how to implement the algorithm (see above), it's not clear whether the Bernoulli setting is just particularly tractable, or whether the same method could in practice be used for other exponential-family distributions.


Additional comments:
- Table 1: the absence of confidence intervals or other uncertainty quantification means we cannot evaluate the claim of "the active mode having a slight edge" (line 308).
- Proposition 2, missing a comma in the first line of math before w_{a,i}^\star.
- Caption of Figure 2, mu is a 3x3 matrix, so it's not immediately clear which dimension is J and which is K. Consider adding $\in [0,1]^{K\times J}$ (or the other way around, whichever it is).


Update 8/16:
I would like to thank the authors for their thoughtful rebuttal, which clarified the questions I had about the algorithm. In particular, in light of the authors' proposed clarifications (the optimization from [6], clarification that this is indeed the suitable approximation mentioned in the text, and discussion on when the GLRT threshold is easy to obtain) I have changed my recommendation to 'accept'. I still think that the presentation of the algorithm in paragraph form is borderline too colloquial for this paper, since the algorithm is an important contribution, and I would encourage the authors to formalize their presentation of the algorithm. However, since code will be provided, I am less concerned about a lack of reproducibility.


**Time Spent Reviewing:**

5

---

> ### Author Response · Authors · 2021-08-10
> **Response to Reviewer x5R6**
>
> We thank the reviewer for his/her detailed assessment of the paper.
>
> ## Question 1:
> The algorithm description did not contain enough detail to reproduce the results in the paper, which is even more problematic because code was not submitted. In particular, line 270 (the generalized likelihood ratio test statistic) is itself an optimization problem, and it is not immediately clear how to generate the threshold. If this is trivial, perhaps include it in the appendix with a reference in the main body. Additionally, I suspect that the phrase "(a suitable approximation of)" in lines 259-260 refers to the iterative saddle point solver mentioned in lines 251-252, but it would be nice to see this made explicit.
>
> **Question 1.1:** [...] did not contain enough detail to reproduce the results in the paper
>
> **Answer 1.1:** We present the essence of the method in lines 255-266, with one explicit simplification, hidden by the 'suitable approximation'. This hides a computational speedup from [6] that we will import in the appendix to make the paper self contained.
>
> **Question 1.2:**
> In particular, line 270 (the generalized likelihood ratio test statistic) is itself an optimization problem
>
> **Answer 1.2:**
> The GLRT is indeed an optimisation problem, which can be solved efficiently for exponential families for which the kl divergence is convex in the second argument, by means of standard convex programming software. This includes the Gaussian, Bernoulli and Poisson cases we mainly focus on in the paper. We will make it explicit in the text.
>
>
> **Question 1.3:**
> [...] it is not immediately clear how to generate the threshold.
>
> **Answer 1.3:**
> In 273 we provide a theoretically safe threshold, but in our experiments we use the more lenient threshold used in [1], which is still very conservative for our experiment setting, as seen in figure 2 (left).
>
> **Question 1.4:** [...] more problematic because code was not submitted
>
> **Answer 1.4:**
> The code was not provided because the repository contained author sensitive information. We will add the link to the code in the final version of the paper. The code implements, besides Bernoulli, the Gaussian and Poisson cases.
>
> ## Question 2:
> After reading the paper, it was unclear whether the method can be practically run for other members of the exponential family.
>
> ## Answer 2:
> The main limitation is the GLRT optimisation problem, which is the core computation both in sampling and stopping rules. We can currently numerically solve it efficiently only for exponential families with KL convex in its second argument. We will add a comment in the paper but this is not only limited to Bernoulli distributions and works also for Poisson distribution or Gaussian with known variance.
>
> ## Other questions
> **Question 3:**
> Table 1: the absence of confidence intervals or other uncertainty quantification means we cannot evaluate the claim of "the active mode having a slight edge" (line 308).
>
> **Answer 3:**
> The CI's on sample complexities over multiple bandit instances are very wide and asymmetric due to the heavy tailed distribution and are often omitted in this line of work (for instance [1], [2]). But as the reviewer suggests, we will add them and remove the claim of “slight edge”.
>
> **Question 4:**
> mu is a 3x3 matrix, so it's not immediately clear which dimension is J and which is K
>
> **Answer 4:**
> Good catch, we will clarify this.
>
> **Limitations and societal impact.**
> We agree with the reviewer that we perhaps answered a bit hastily on this point. In the experiment reported in the paper the subpopulations correspond to time slots, which is used to model a form of seasonality in the user responses. This use of the model does not raise (we believe) questions in terms of societal impact. On the other hand, if the subpopulations are formed, say, based on user individual characteristics then the impact might be different and we agree that the definition of the best overall arm (that is the choice of the beta vector) should be considered carefully.
>
> **References**
>
> [1] Garivier et al., "Optimal Best Arm Identification with Fixed Confidence", COLT 2016
>
> [2] Kaufmann, E., & Kalyanakrishnan, S. Information complexity in bandit subset selection. In Conference on Learning Theory (pp. 228-251). PMLR.

---

> > ### Comment · Reviewer_x5R6 · 2021-08-16
> > **No CI's needed for Table 1 if "slight edge" claim is removed**
> >
> > Re: Q3 - you don't have to include confidence intervals, if you remove the 'slight edge' claim. My only point was that the claim isn't backed up by the data currently in the table, and I would need some idea of the confidence intervals if I were to believe it.
> >
> > Also, please see my updated official review for the response to the rest of your rebuttal.

---

### Official Review · Reviewer_VNse · 2021-07-18

**Rating:** 5
**Confidence:** 3

**Summary:**

In this paper, the authors develop and analyze several variants of a bandits algorithm that takes into account possible different subpopulations, and employs an adaptive stopping time.

**Limitations And Societal Impact:**

.

**Main Review:**

In this paper, the authors develop and analyze several variants of a bandits algorithm that takes into account possible different subpopulations, and employs an adaptive stopping time.
The paper derives theoretical results on the sample complexity of the proposed procedure, along with a valid "risk assessment" strategy.

Overall, the paper is clearly written and well motivated. There are some deficiencies, which may be more generally related to the background literature rather than the particular paper --- I would appreciate the authors' feedback on that. Specifically:

* It seems too restrictive to assume the same model for all different subpopulations. Is this common in the literature?

* Moreover, it seems very restrictive to assume the same one-parameter exponential family model.
  (a) Why only one parameter ? (What about the case of normal with mean \mu_{a, i} and variance \sigma^2_{a, i} as in Prop 2? Do we assume known variances here?
   (b) Why exponential family? It is clear that this introduces several simplifications as the log-likelihood is linear in the parameter, but this is not discussed in the paper. What specific structure do we need from the exponential family? How plausible is this for real-world applications?

* What does kl(p, q) look like? It would help to have an idea about how this depends on, say, |p-q|.

* Prop 2 was useful. Is it obvious how to order the T^* from various settings?

* The sampling rules from L255 and the recommendation in L268 are not entirely clear to me. Would it be possible to present in an Algorithm format?

* The application in 4.2 involves time, which is a mismatch with the theory in the paper (which focused on subpopulations).
So, I am not sure how much this adds to the paper.







**Time Spent Reviewing:**

5

---

> ### Author Response · Authors · 2021-08-10
> **Response to Reviewer VNse**
>
> Thank you for your insightful comments.
>
> ## Question 1:
> - It seems too restrictive to assume the same model for all different subpopulations. Is this common in the literature?
> - Moreover, it seems very restrictive to assume the same one-parameter exponential family model. (a) Why only one parameter ? (What about the case of normal with mean \mu_{a, i} and variance \sigma^2_{a, i} as in Prop 2? Do we assume known variances here? (b) Why exponential family? It is clear that this introduces several simplifications as the log-likelihood is linear in the parameter, but this is not discussed in the paper. What specific structure do we need from the exponential family? How plausible is this for real-world applications?
>
> ## Answer 1:
> Assuming the same model for different subpopulations does not seem to be a strong assumption. For an online advertising company that focuses on the Click-Through-Rate (CTR), the conversion rate can be modelled with Bernoulli distributions. Note that for binary outcomes EVERY distribution is Bernoulli, so the model is always well-specified and hence plausible for the real-world. While different subpopulations could have different conversion rates, it seems reasonable to use Bernoulli distribution with a different mean for different subpopulations. This case is covered by our model where Bernoulli distributions with different means depending on the subpopulations can be used. Furthermore, most of frequently used distributions are included in a one-parameter exponential family (Bernoulli, Poisson, Gaussian distribution with known variance). A lot of existing algorithms are analyzed under the same assumption, in the pure exploration setting ([1,2]) or in the regret minimisation setting ([3,4]) (just to name a few) and we do not think that this aspect is a limiting factor of our work.
>
> ## Question 2:
>
> The application in 4.2 involves time, which is a mismatch with the theory in the paper (which focused on subpopulations).
>
> ## Answer 2:
>
> This is incorrect, and in fact we designed this experiment on purpose to illustrate why the method we develop is general. The different times of the day can be grouped, we used 6-hours consecutive period as explained on line 330. In this experiment, the "subpopulations" are the 4 quarters of the days. The proposed method works not only for categorical variables, but can also be used for temporal data where there is a form of non-stationarity.
>
> ## Other questions
> **Question 3:** What does kl(p, q) look like? It would help to have an idea about how this depends on, say, |p-q|.
>
> **Answer 3:** Pinsker's inequality states that kl(p,q)>=2(p-q)^2, see e.g. Lemma 10.2 in [5].
>
> **Question 4:** Prop 2 was useful. Is it obvious how to order the T^* from various settings?
>
> **Answer 4:** This is the purpose of Equation 6 in the main paper.
>
> **Question 5:**
> The sampling rules from L255 and the recommendation in L268 are not entirely clear to me. Would it be possible to present in an Algorithm format?
>
> **Answer 5 :**
> We will add the pseudocode of the algorithm in the Appendix.
>
>
>
> ## References
>
> [1] Garivier et al., "Optimal Best Arm Identification with Fixed Confidence", COLT 2016
>
> [2] Degenne et al., "Pure Exploration with Multiple Correct Answers", NeurIPS 2019
>
> [3] Cappe et al., "Kullback-Leibler upper confidence bounds for optimal sequential allocation", The Annals of Statistics 2013
>
> [4] Baudry et al., "Sub-sampling for Efficient Non-Parametric Bandit Exploration", NeurIPS 2020.
>
> [5] Lattimore et al, "Bandits Algorithms", Cambridge Press.
>
>
> All in all we think Reviewer 2 overlooked crucial aspects of our work and we encourage to reevaluate the appreciation of the paper given the proposed answers.

---

### Official Review · Reviewer_KZHw · 2021-07-20

**Rating:** 5
**Confidence:** 2

**Summary:**

Paper consider a/b/n testing into a bandit problem. Paper raised a framework and give an algorithm which reaches the complexity lower bound. Paper provided simulations showing advantages of the proposed algorithm.

**Limitations And Societal Impact:**

No. Theoretical work so the potential societal impact is limited. Potentially will have applications on social network data, etc. So might be good to mention this.

**Main Review:**

(not an expert in the area so will provide general reviews. please refer to other reviewers on evaluation on the tech details)
Paper provided a solution to the A/B/n testing problem which is very useful in the industry data analysis
- Paper constructed a complexity bound and provided algorithm reaching it, constructing an optimal framework
- Simulation showing superior performance
- Various parameter models are analysed and they are widely applicable to the real setups

However the paper is hard to follow, especially since there is no illustrations of algorithm/ toy example for problem setup to understand the math setup, etc. The presentation is a bit packed (might because of the page limit). It might be good to focus on a sub-problem, such as most profitable arm identification / AB testing with 2 subpopulations and present the general setup results as an extension.

**Time Spent Reviewing:**

4

---

> ### Author Response · Authors · 2021-08-10
> **Response to Reviewer nYMu**
>
> Thank you for your comments. We have already included some easy cases to simplify the problem (see Section 2.4 and 2.5). In particular, Section 2.5 considers the Gaussian case with 2 arms and different subpopulations: we give explicit formulas for the weights and the characteristic times of the problem. Those are the key characteristics for understanding the difficulty of the learning objective: identifying the set of arms that are better than the control arm (whose performance is unknown).
>
> The pseudo-code of the algorithm will be added in the Appendix to clarify this.

---

### Decision · Program_Chairs · 2021-09-27

**Decision:**

Accept (Poster)

**Comment:**

Dear authors,

There was some disagreement between the reviewers.
Given this, and taking into account all comments, as well as background of the reviewers, I decided to take a special look at the article.

After reading the article, I believe this setting is well-motivated and interesting.
The lower bounds do not seem very hard to derive, but I appreciate the effort to make them explicit on an illustrative example.
The proof of Theorem 2 is heavily relying on previous work, working by reduction.
Clarity should indeed be improved, regarding presentation of the algorithm and notations.
[Small point: I suggest you add $\\beta$ in the notation for $S(\\mu)$ or $Alt(\\mu)$, and otherwise it is not clear where is the dependency in beta in Theorem 1]
Regarding the algorithm, I do not encourage using a different beta(t,\delta) for the analysis to be true and a "stylized one" for the experiments, as this indicates mismatch between theory and experiments.
Now, the algorithm does achieve optimality in the considered setting and is supported by illustrative numerical results.

The discussion of the complexity of the algorithm and the handling of the $2^K$ sets to consider should be improved.
Proposition 3 brings an interesting point, but I feel this part could be extended (for instance, what happens then for other distributions, is there any hope of having some reasonable complexity?)
The numerical experiments should also be clarified, and seem to be restricted to small values of K: I encourage to consider examples with much larger $K$ as well, and perhaps highlight the (low?) computational complexity of the algorithm.


All in all, I believe this is a simple adaptation of an existing work to a relevant problem. The paper has interesting bits, but each of them seem done shallowly. This makes the paper unfortunately borderline. I recommend acceptance, conditioning on the fact the authors provide more convincing discussion and experiments regarding the numerical complexity, but only if there is enough place for that.
Otherwise I would be happy to see an updated version of this article taking into accounts all feedbacks.